# Hydrogen trapping and embrittlement in high-strength Al alloys

Huan Zhao[1✉], Poulami Chakraborty[1], Dirk Ponge[1], Tilmann Hickel[1,2], Binhan Sun[1,3], Chun-Hung Wu[1], Baptiste Gault[1,4✉] & Dierk Raabe[1✉]

Ever more stringent regulations on greenhouse gas emissions from transportation motivate efforts to revisit materials used for vehicles[1]. High-strength aluminium alloys often used in aircrafts could help reduce the weight of automobiles, but are susceptible to environmental degradation[2,3]. Hydrogen 'embrittlement' is often indicated as the main culprit[4]; however, the exact mechanisms underpinning failure are not precisely known: atomic-scale analysis of H inside an alloy remains a challenge, and this prevents deploying alloy design strategies to enhance the durability of the materials. Here we performed near-atomic-scale analysis of H trapped in second-phase particles and at grain boundaries in a high-strength 7xxx Al alloy. We used these observations to guide atomistic ab initio calculations, which show that the co-segregation of alloying elements and H favours grain boundary decohesion, and the strong partitioning of H into the second-phase particles removes solute H from the matrix, hence preventing H embrittlement. Our insights further advance the mechanistic understanding of H-assisted embrittlement in Al alloys, emphasizing the role of H traps in minimizing cracking and guiding new alloy design.

High-strength Al alloys of the 7xxx series are essential structural materials in aerospace, manufacturing, transportation and mobile communication[5,6], owing to their high strength-to-weight ratio, which enables products with lower fuel consumption and environmental impact. The high strength is achieved through the formation of a high number density (approximately $10^{24}$ m$^{-3}$) of nanosized precipitates via an aging thermal treatment[6,7]. Unfortunately, high-strength Al alloys are notoriously sensitive to environmentally assisted cracking[2,8], and, like most high-strength materials, are subject to H embrittlement[9,10]. Overcoming these intrinsic limitations requires gaining a precise understanding of how H penetrates the material and of its interactions with ubiquitous microstructural features, for example, grain boundaries (GBs) or second phases, to ultimately cause a catastrophic deterioration of mechanical properties[11]. H uptake can occur during high-temperature heat treatments, as well as in service[12,13]. H has low solubility in Al[14], yet crystal defects can assist H absorption[15–22], leading, for instance, to a drop in the fatigue life[23].

The enduring question remains of where the H is located in the microstructure and how such traps facilitate catastrophic failure. Several studies pointed to the critical role of GBs in the environmental degradation. GBs are locations of preferential electrochemical attack[4], but also cracks propagate more easily via GB networks throughout the microstructure of the alloy[24,25]. An experimental validation of the H distribution in Al alloys is challenging, owing to its low solubility and to the experimental difficulty of performing spatially resolved characterization of H at relevant scales and at specific microstructural features. Recent efforts in atomic-scale H imaging in steels led to insights into the trapping behaviour of second phases and interfaces[26–28].

Here we address these critical questions using the latest developments in cryo-atom probe tomography (APT)[26–28], enabled by cryo-plasma focused-ion beam (PFIB) specimen preparation to investigate H associated with different microstructures in an Al alloy. Through isotope-labelling with deuterium (D), we partly avoid characterization artefacts associated with the introduction of H from the sample preparation[28,29] and from residual gas in the atom probe vacuum chamber. We studied a 7xxx Al alloy with a composition of Al–6.22Zn–2.46Mg–2.13Cu–0.155Zr (wt.%) in its peak-aged condition. For this alloy, electrochemical-charging with H leads to a critical drop in the ductility compared with uncharged samples (Fig. 1a). The H desorption spectra are shown in Extended Data Fig. 1. Figure 1b–d highlights the complexity of the microstructure across multiple length scales. First, Fig. 1b, c reveals the predominant role of GBs and GB networks in the crack formation and propagation during deformation of the H-charged material. Fig. 1d displays the typical distribution of fine precipitates in the bulk, coarse precipitates at GBs and precipitate-free zones (PFZs) adjacent to GBs. Intermetallic phases (for example, the Al$_2$CuMg S phase) and Al$_3$Zr dispersoids that act as grain refiners are also shown.

Peak-aged specimens were electrochemically charged with D for subsequent APT analysis after validating that H and D show a similar embrittling effect on mechanical properties (Extended Data Fig. 2). D-charged specimens were prepared by PFIB at cryogenic temperatures to limit the introduction of H[29], and immediately analysed by APT using voltage pulsing to minimize residual H from APT[28,29]. Figure 2a displays the APT analysis of Al$_3$Zr dispersoids in the D-charged specimen, with the corresponding composition profile shown in Fig. 2b. H is strongly enriched, up to 9.5 at.% on average, within the dispersoids,

[1]Max-Planck-Institut für Eisenforschung, Düsseldorf, Germany. [2]BAM Federal Institute for Materials Research and Testing, Berlin, Germany. [3]Key Laboratory of Pressure Systems and Safety, Ministry of Education, School of Mechanical and Power Engineering, East China University of Science and Technology, Shanghai, China. [4]Department of Materials, Royal School of Mines, Imperial College London, London, UK. ✉e-mail: h.zhao@mpie.de; b.gault@mpie.de; d.raabe@mpie.de

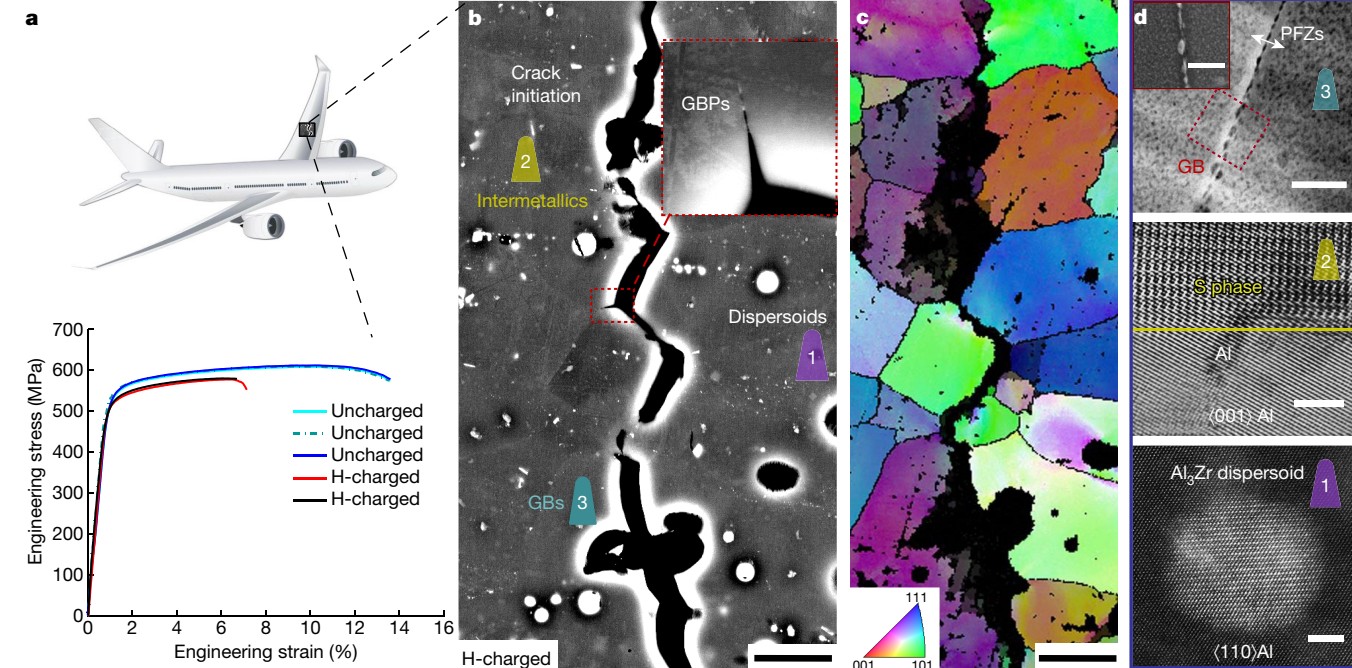

**Fig. 1 | Heterogeneous microstructure of an aerospace Al–Zn–Mg–Cu alloy.** **a**, Engineering stress–strain curves of uncharged and H-charged samples in the peak-aged condition (120 °C for 24 h). **b**, Backscattered electron imaging of an intergranular crack of the H-charged alloy subjected to tensile fracture. **c**, Electron backscatter diffraction imaging showing the crack along GBs. **d**, The microstructure of GBs, precipitates, PFZs[31] and main types of secondary phases (the S phase[47] and $Al_3Zr$ dispersoid). The colour schemes reflect the microstructures where specific APT analyses were performed. APT, atom probe tomography; GB, grain boundary; GBPs, grain boundary precipitates; PFZs, precipitate-free zones. Scale bars: 20 μm (**b**, **c**), 100 nm (**d**, top), 50 nm (**d**, top inset), 3 nm (**d**, middle and bottom).

contrasting with the much lower content of only 0.4 at.% H in the Al matrix. D is also enriched up to 2.8 at.% inside the dispersoids. H and D atoms partition preferably to sites inside the dispersoids, with a slightly higher content at the interface that may be due to the misfit strain (0.75%)[30]. We further analysed uncharged specimens prepared by PFIB and electrochemical polishing for reference (Extended Data Fig. 3). H appears consistently enriched inside $Al_3Zr$ dispersoids, up to 8.5 at.% on average. Only a peak at a mass-to-charge ratio at 1 Da, corresponding to $H^+$ atomic ions, is detected in the dispersoids in uncharged specimens. However, in the D-charged material, a distinct peak at 2 Da gives proof of efficient D-charging, with D partitioning into $Al_3Zr$ dispersoids.

Figure 2c shows the APT analysis on an intermetallic particle in the D-charged sample. The composition profile indicates that the Mg-enriched region corresponds to the S phase ($Al_2CuMg$). The S-particle contains 4.2 at.% H, whereas the matrix has only 0.3 at.% H, and 0.12 at.% D (right axis). Comparison with a similar S particle in an uncharged sample (Extended Data Fig. 4) shows a 6.5-times higher peak ratio of 2 Da/1 Da in the D-charged sample, revealing that most of the signal at 2 Da comes from electrochemically charged D. Further evidence of an enrichment up to 9 at.% H within $Al_7Cu_2Fe$, and $Mg_{32}(Zn,Al)_{49}$ T-phase particles, is provided for the uncharged material (Supplementary Figs. 1, 2).

We then analysed the distribution of H and D at a high-angle GB. Following sharpening at cryo-temperature, the specimen was transferred through a cryo-suitcase into the APT to minimize out-diffusion of D[28]. The peak-aged sample contains 5-nm (Mg, Zn)-rich strengthening precipitates in the bulk and coarser 20–50-nm-sized precipitates at the GB[31], as well as typical PFZs adjacent to the GB (Fig. 3a). Atom maps of H and D($H_2^+$) in Fig. 3b reveal a higher concentration at the GB. Fig. 3c shows details of the precipitates and solutes populating the GB. $Al_3Zr$ dispersoids at the GB (Fig. 3d) contain 11 at.% H and 0.6 at.% D—that is, a lower D content compared to the $Al_3Zr$ dispersoids in the bulk (Fig. 2b).

No enrichment in H and D($H_2^+$) (right axis) is shown in (Mg, Zn)-rich precipitates distributed both at the GB (Fig. 3e) and in the bulk (Extended Data Fig. 5). Fig. 3f gives a composition profile through the GB between the particles, showing that the GB is enriched with 2 at.% Mg. We observe no enrichment in Zn and Cu (1 at.%, Extended Data Fig. 6), and in the peak-aged state this can be explained by the accelerated GB precipitation through the consumption of segregated solutes[31]. The locally increased content of D($H_2^+$) implies that the solute-decorated GB (that is, devoid of precipitates) acts as a trap for H, and no enrichment in H and D is observed in the adjacent PFZs (that is, the regions next to the GB), an effect that amplifies the mechanical and electrochemical contrast in these regions. Comparison with a similar GB in an uncharged sample (Extended Data Fig. 7) shows a higher signal at 2 Da (by a factor of 3) in the D-charged sample, supporting that D is indeed enriched at the GB. We obtained seven APT datasets containing GBs in D-charged samples, and all show consistent enrichment of H and D at GBs (two additional datasets are shown in Supplementary Figs. 3, 4).

We note that the probability of detecting spurious H from residual gas in APT decreases as the strength of the electric field increases, which can be traced by the evolution of the charge-state ratio of Al (that is, $Al^{2+}/Al^{1+}$)[32]. For each microstructural feature studied herein, this ratio is reported in Extended Data Fig. 8, and in each H-enriched case, the electric field either does not change notably or increases compared to Al matrix. The content of H and D measured in each feature in the uncharged and D-charged conditions is compiled in Supplementary Table 1. These analyses indicate that the peak at 2 Da is extremely unlikely to be associated with $H_2^+$, but with D in D-charged samples, and that most of the detected H was from initially trapped atoms inside the specimen, either from its preparation or/and from the processing history of the material[28]. The electrolytical-charging with D reinforces our observation that H is trapped within the material itself[28].

To better understand the effect of H in the intermetallic phases (for example, S phase $Al_2CuMg$), $Al_3Zr$ dispersoids and at GBs, we

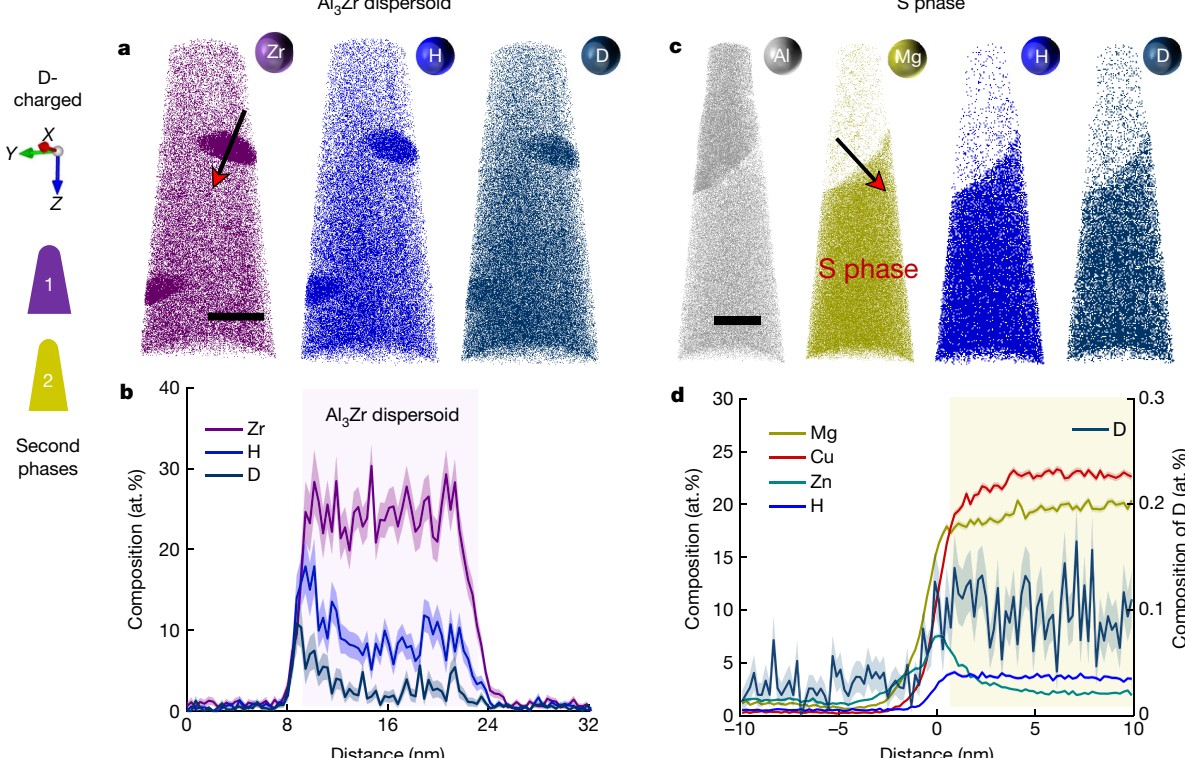

**Fig. 2 | APT analysis of second phases of the D-charged Al–Zn–Mg–Cu samples in the peak-aged condition (120 °C for 24 h). a–d**, Atom map and composition profiles are presented along the red arrows respectively for Al₃Zr dispersoids (**a**, **b**) and S phase (**c**, **d**). The shaded bands of the traces correspond to the standard deviations of the counting statistics in each bin of the profile. The background colours in **b**, **d** show the locations of the dispersoid and the S phase, respectively. Scale bars: 30 nm (**a**, **c**).

used density functional theory (DFT). Solubility analysis of H in the S phase reveals that Al-rich octahedral sites provide the lowest solution enthalpy (0.014 eV). The calculated concentrations of H in these sites is 3 at.% at 300 K, substantially higher than $5 \times 10^{-5}$ at.% assumed for the Al matrix, which explains the APT observations. In Al₃Zr dispersoids, H prefers octahedral interstitial sites with Zr in the second nearest-neighbour shell and with a solution enthalpy of 0.128 eV and a H solubility of 0.2 at.%. However, the high experimental H concentrations may be explained by the presence of Zr antisites in the first nearest-neighbour positions of H, which reduces the solution enthalpy to −0.202 eV. The solubility of H in the GB was estimated for a symmetric Σ5 (210) [100] GB (Fig. 4a) as a representative high-angle GB[33]. The excess volume for all considered GB sites (Fig. 4b) explains the negative segregation energies given in Fig. 4c. Therefore, the corresponding solution enthalpies at these sites are lower than in the Al matrix, but still much higher than in the S phase or Al₃Zr dispersoids.

To explain why GBs, nevertheless, show higher susceptibility for H embrittlement, as documented in Fig. 1, we determine the embrittling energy associated with H (Fig. 4c). This quantity describes the thermodynamic driving force for fracture formation by comparing the impact of H on the energetics of the GB with that of the free surface. In the Σ5 GB, H when located at sites with the strongest segregation energy, also yields the strongest embrittlement. When distributing H atoms according to their nominal solubility over all these possible sites in a unit area of the GB, weighted by their respective segregation energy (Fig. 4c), the total contribution to the embrittling energy adds up to 600 mJ m⁻² for this GB. This value is substantially more positive (that is, more detrimental) than the embrittling energy determined for Al₃Zr dispersoids (2 mJ m⁻²) and the S phase (129 mJ m⁻²).

We investigate the effect of Mg segregation on GBs revealed by APT and introduce in the simulation cell a Mg atom substituting one of four equivalent Al atoms in the GB plane (Fig. 4d). The negative segregation energy of Mg (−0.274 eV) indicates that it stabilizes the GB compared to defect-free Al[34,35], whereas the small negative embrittling energy (−0.043 eV) yields almost no effect on the GB strength compared to the formation of free surfaces. However, for H added to the GB supercell into the interstitial sites at and near the segregated Mg atom (Fig. 4a), the embrittling energy changes greatly, as summarized in Fig. 4c. The solution enthalpy gives no indication that co-segregation of Mg and H is energetically favourable. In particular, H sitting at the capped trigonal prisms maintains its strong (that is, negative) segregation energy and has a strong (positive) embrittling energy that is considerably enhanced in the presence of Mg. In the same way, all other sites substantially contribute to embrittlement when a Mg atom is nearby and when H diffusion at the opening free surface is considered. This is even true for sites such as 1i and 1gb, for which an occupation by H is less probable. Overall, these effects increase the embrittling energy by H per unit GB area by approximately one order of magnitude with Mg compared to the Mg-free case. The opposite impact of Mg on the segregation and the embrittlement caused by H is explained by the interaction of Mg and H at the free surface resulting from the decohesion.

We can now rationalize the H-embrittlement mechanism as follows: as H penetrates the alloy, the equilibrium H concentration remains low in the Al matrix and also in the fine and coarse (Mg, Zn)-rich precipitates. However, H accumulates in intermetallic phases (for example, S or T phases), Al₃Zr dispersoids, and to a lesser extent, at GBs. The high H enrichment in the second-phase particles was explained by DFT calculations where H shows no clear decohesion or embrittlement effects. Upon H saturation of the second phases, further ingress of H will gradually lead to an accumulation of H at GBs. DFT predicts no strong increase in H concentrations in the presence of Mg, which agrees with APT where H is not strongly segregated at GBs compared to second

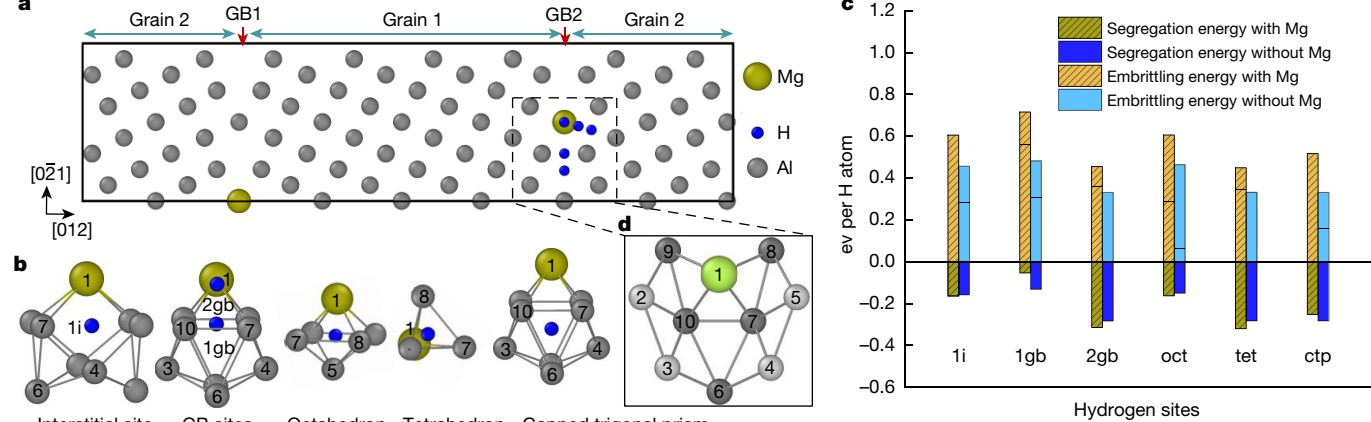

**Fig. 3 | APT analysis of a D-charged peak-aged Al–Zn–Mg–Cu sample containing a GB (120 °C for 24 h). a**, The iso-surfaces highlight the dispersion of fine (Mg, Zn)-rich precipitates in the matrix, coarse ones at the GB, and $Al_3Zr$ dispersoids. Scale bar: 30 nm. **b**, Atom maps of H and D($H_2^+$). **c**, Solute distribution at the GB plane. **d**, Composition profile across one $Al_3Zr$ dispersoid at the GB. **e**, Composition profile of one (Mg, Zn)-rich precipitate at the GB. **f**, Solute composition profile across the GB in between precipitates. The shaded bands of the traces correspond to the standard deviations of the counting statistics in each bin of the profile.

phases. Yet DFT calculations suggest that when combined with Mg, the strong driving force for H to segregate to a free surface with respect to a possible interstitial site at GBs favours GB decohesion and drives the

system towards crack formation. This rationalizes that GBs are embrittled and explains that Mg can impact the H embrittlement without promoting the absorption of H to GBs[11,36]. Further investigation on the

**Fig. 4 | Theoretical analysis based on DFT simulations. a**, Schematic representation of the symmetric Σ5 (210) GB in Al shown with two GB planes. **b**, The projected and perspective views of deltahedral packing units show the H adsorption sites of the calculations. Site number 1 is the substitutional site for a Mg atom nearby the H sites located inside the polyhedral packing units. **c**, The

embrittling energy and segregation energy are compared in the absence and presence (patterned bars) of Mg as a solute atom at the GB for the different interstitial sites of H labelled in **b**. **d**, The Al (light grey) and Mg atoms (light green) in the enlarged figure belong to different adjacent (001) planes.

elemental distribution at a H-induced intergranular crack using scanning Auger electron microscopy reveals the enrichment of Mg at the cracked GB (Extended Data Fig. 9). The enrichment is even stronger (by a factor of 2) than the Mg concentration at the GB (Fig. 3f). To confirm the generality of the role of Mg we also show that no H-embrittlement features occurred in a Mg-free Al−5.41 (wt.%) Zn alloy that was used as reference material. The alloy was exposed to the same H-charging and tensile test conditions, but no sign of H embrittlement was found, neither in the tensile test results nor in the metallographic fractography (Extended Data Fig. 10). These findings support the result that the co-segregation of Mg and H to free surfaces provides the driving force for the embrittlement of GBs.

Generally, avoiding the ingress of H in the first place is extremely unlikely to work, and the best approach to mitigate H embrittlement is therefore to control its trapping to maximize the in-service lifetime of the components. Our results provide indications of H-trapping sites and their respective propensity to initiate damage in environmentally assisted degradation, thus contributing towards establishing a mechanistic understanding of H embrittlement in Al alloys. On this basis of this study, we propose specific measures that may be explored to enhance resistance to H-induced damage and improve the lifetime and sustainability of high-strength lightweight engineering components. The results on the high H enrichment in second-phase particles provide a potential mitigation strategy for improving H-embrittlement resistance, namely through introduction and manipulation of the volume fraction, dispersion and chemical composition of the second phases, despite their potentially harmful effects on mechanical properties. Other strategies could aim at designing and controlling GB segregation, for instance with the goal of eliminating Mg decoration of GBs by trapping it into precipitates and keeping it in bulk solution. A third and more general avenue against environmental degradation lies in reducing the size of PFZs in these alloys, with the goal to mitigate the H-enhanced contrast in mechanical and electrochemical response between the H-decorated GBs and the less H-affected adjacent regions.

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

## Methods

### Materials

The chemical composition of the Al alloy studied is Al−6.22Zn−2.46Mg−2.13Cu−0.16Zr−0.02Fe in wt.% (Al−2.69Zn−2.87Mg−0.95Cu−0.05Zr−0.01Fe in at.%). The as-cast ingot was homogenized at 475 °C and water quenched followed by hot rolling at 450 °C to 3 mm thickness. Samples were cut to the size of 12 mm × 10 mm × 2 mm and then they were solution treated at 475 °C for 1 h and quenched in water. Peak aging was immediately carried out by heat treatment at 120 °C for 24 h with water quenching. The detailed precipitation behaviour during aging is described in a previous work[7].

An Al−5.41Zn−0.15Fe−0.02Si in wt.% (Al−2.31−0.08Fe−0.02Si in at.%) alloy was used as reference material, which contains a similar amount of Zn as in the studied Al−Zn−Mg−Cu alloy. The cast ingot was homogenized at 360 °C for 6 h and water quenched, followed by hot rolling at 345 °C from 20 to 3 mm thickness and solution treated at 360 °C for 1 h and a final quench in water.

### Microstructure observations

The microstructures of the cracks and the adjacent regions were characterized by the combined use of backscattered electron imaging (Zeiss-Merlin SEM) and electron backscatter diffraction (Sigma). For transmission electron microscopy (TEM), specimens were prepared by in situ lift-out, using a dual-beam PFIB instrument. The microstructures of specimens prepared for TEM probing were analysed using a JEOL-2200FS operated at 200 kV or an aberration-corrected FEI Titan Themis 80−300 operated at 300 kV. Auger analysis was performed on a JEOL JAMP 9500 F Auger spectrometer with a cylindrical mirror analyser and a thermal emission electron gun. The operating vacuum pressure of the chamber was about $5 \times 10^{-7}$ Pa. The accelerating voltage ($E_p$) of the electron beam is 25 kV and the probe current ($I_p$) is about 10 nA, the Auger measurements were conducted at a working distance 23.2 mm, with the sample being tilt by 30°. Before the mapping started, the sample was pre-sputtered to remove surface contaminations. The scanning energy intervals of each element—O (495.6−518.4 eV), Al (1,453.2−1,504 eV), Mg (1,175.0−1,188.0 eV), Cu (896.0−930.0 eV), Zn (970.0−1,004.0 eV)—and the mapping settings (dwell time, 50 μs, number of accumulations, 10) were identical for all elements. The intensity definition of the obtained map is $(P − B)/B$ ($P$, peak, $B$, background).

### Deuterium charging method

Deuterium (D) charging was conducted on a three-electrode electrochemical cell as shown in a previous work[28]. A charging solution of 0.05M NaCl with 0.03 wt.% $NH_4SCN$ in $D_2O$ (Sigma-Aldrich) was used as the cathode electrolysis to create a D-rich environment around the Al samples. A platinum counter-electrode and reference (μ-Ag/AgCl) were used. The D charging was conducted for 3 days to 1 week, followed by immediately transferring the samples to PFIB. For all charging experiments, a PalmSens EmStat3 potentiostat was used.

### TDS measurements

Thermal desorption spectroscopy (TDS) experiments were performed using a Hiden TPD Workstation to measure the H concentration in both H-charged and uncharged reference specimens. Specimens with a dimension of $10 \times 15 \times 1.0$ mm$^3$ were used, and the TDS spectra were measured at a constant heating rate of 16 °C min$^{-1}$. Three samples were measured for each condition in the H-charged and uncharged state. The charging was conducted on a three-electrode electrochemical cell for 3 days. A charging solution of 0.05M NaCl with 0.03 wt.% $NH_4SCN$ in $H_2O$ was used, after which the tests were started within 15 min. The total H concentration was determined by measuring the cumulative desorbed H from room temperature to 400 °C.

### Tensile experiments

Tensile testing was conducted on a Kammrath & Weiss test stage coupled with the digital image correlation (DIC) technique. Tensile specimens with a gauge length of 8 mm and a width of 2 mm were used. The tests were performed at a strain rate of $3 \times 10^{-4}$ s$^{-1}$. At least five samples were tested for each condition (uncharged, H-charged and D-charged). Global and local strain distributions were measured by DIC. The data analysis was done using the ARAMIS software.

### APT sample preparation

For the APT specimens prepared by electrochemical polishing, samples were first cut into 1 mm × 1 mm × 12 mm bars. First rough polishing was conducted in a solution of 25% perchloric acid in glacial acetic acid at 10−30 V. Final polishing was done in 2% perchloric acid in 2-butoxyethanol under an optical microscope. For the APT specimens prepared by PFIB, bulk samples with the size of 10 mm × 12 mm × 1 mm were prepared on an FEI Helios PFIB instrument operated with a Xe source to avoid contamination by gallium.

For the APT specimens prepared from grain boundaries (GBs), GBs were first crystallographically characterized and then site-selected in the SEM, and trenches were cut from the GBs in the plate samples. D charging was then conducted in the bulk plate samples. After charging, the samples were immediately transferred to PFIB, lifted out from the pre-cut trenches, and mounted to the Si coupons. The sharpening processes were done at a cryo-stage fitted with a Dewar and a cold finger. More details on this specific setup can be found in previous works[28,29,37]. The cryo-prepared APT specimens were transferred from the PFIB into APT under cryogenic ultrahigh vacuum (UHV) conditions using our cryogenic UHV sample transfer protocols described previously[37].

### APT experiments

Atom probe measurements were performed on the local electrode atom probe (LEAP 5000XS/LEAP 5000XR) at a cryogenic temperature of 25 K under UHV conditions of $10^{-11}$ Torr. All APT measurements were carried out using voltage pulsing with a 20% pulse fraction and a 250 kHz pulse rate. Multiple APT datasets were obtained from multiple APT tips prepared from GBs and second-phase particles. APT datasets were analysed using the commercial software package IVAS 3.8.4. The APT reconstruction parameters were calibrated according to the crystallographic poles appearing on the detector hit maps[38].

### Computational details

The DFT calculations were carried out using the projector augmented wave (PAW) potentials as implemented in VASP[39,40]. The exchange and correlation terms were described by the generalized gradient approximation (GGA) proposed by Perdew, Burke and Ernzerhof (PBE)[41]. A plane-wave cut-off of 500 eV was taken for all calculations. The convergence tolerance of atomic forces is 0.01 eV Å$^{-1}$ and of total energies is $10^{-6}$ eV. The $k$-point sampling number was set large enough that the convergence of the total energies was within 2 meV per atom. Brillouin zone integration was made using Methfessel−Paxton smearing. Ionic relaxations were allowed in all calculations keeping the shape and volume fixed. The equilibrium structure for pure Al with a lattice parameter of 4.04 Å obtained within the convergence criteria is consistent with previous DFT-GGA calculations[42] and has been used to construct the supercells.

The H solubility across microstructural features, denoted as $\sigma$, can be calculated as:

$$c_H = \exp[-H_{sol}^{\sigma}(H)/k_B T]$$
$$= \exp\left[-(E_{AlX+H}^{\sigma} - E_{AlX}^{\sigma} - \mu_H)/k_B T\right],$$

where $E_{AlX+H}^{\sigma}$ is the total DFT energy of the supercell, $H_{sol}^{\sigma}(H)$ is the solution enthalpy of H in the phase $\sigma$, $X$ is an impurity as explained in

the next section, and $k_B$ is the Boltzmann constant. The chemical potential $\mu_H$ is aligned such that a nominal solubility of ~$5 \times 10^{-5}$ at.% is obtained at $T = 300$ K for the preferred tetrahedral interstitial positions in the face-centred cubic (fcc) Al matrix[43,44]. A $2 \times 2 \times 4$ simulation cell is considered for $Al_2CuMg$ (256 atoms per cell) and the solution enthalpy of H is compared for all possible interstitial sites. $Al_3Zr$ has a $L1_2$ structure with Al atoms at the fcc positions. A $3 \times 3 \times 3$ cell is considered here with 108 atoms in total. The solution enthalpy of H is calculated for the two different octahedral sites in $Al_3Zr$. For each microstructure a consistent simulation cell is considered for bulk and the free surface. The free surface depicts the supercell after crack formation, thereby containing half the number of bulk atoms.

The $\Sigma5$ (210)[100] symmetric tilt grain boundary (STGB) is selected as a representative high-angle GB[33]. The supercell shown in Fig. 4a contains 40 atomic layers (4 atoms per layer, 160 atoms per cell) and represents a cell doubled along the [100] and [012] directions. The GB supercell includes two GBs where two Mg solute atoms are placed at the GB layer such that possible interactions between them are avoided. The free-surface supercell has exactly half the number of atoms, but the same dimensions as the GB supercell and the two Mg solute atoms are placed in symmetrically equivalent positions. The Mg atoms replace one of four equivalent host atoms in the GB plane and the H atom is inserted between host atoms and close to the substituted Mg atom in the GB2 plane as shown in Fig. 4. The dimension of all the models was fixed during structural optimizations, allowing relaxations only along the direction perpendicular to the GB plane. The $2 \times 9 \times 9$ Monkhorst–Pack $k$-point mesh is used in all calculations of GB. All structures have been rendered using the OVITO[45] programme package and all GB structures were created using the software GB Code[46].

## GB segregation

The ability of an impurity $X$ to segregate to the GB can be characterized by the segregation energy given by,

$$E_{seg}^{GB} = (E_{Al+X}^{GB} - E_{Al}^{GB}) - (E_{Al+X}^{bulk} - E_{Al}^{bulk})$$

Here, $E_{Al}^{GB}$, $E_{Al+X}^{GB}$, $E_{Al}^{bulk}$ and $E_{Al+X}^{bulk}$ are the total energy of the pure Al GB, GB in presence of impurity atoms $X$ = Mg or H, pure bulk Al and bulk Al with impurity atoms $X$, respectively. A negative segregation energy indicates that the impurity atoms prefer to segregate towards GB from the bulk environment.

## GB embrittlement

The changes in the mechanical strength of the GB with segregation of impurity atoms is characterized by the embrittling energy $\eta$ within the framework of Rice–Thomson–Wang approach[34,35]

$$\eta = E_{seg}^{GB} - E_{seg}^{FS} = (E_{Al+X}^{GB} - E_{Al}^{GB}) - (E_{Al+X}^{FS} - E_{Al}^{FS})$$

Here free-surface energies (FS) are defined similarly to the corresponding GB energies. A negative value of embrittling energy suggests that the impurity will enhance the GB strength, whereas a positive value indicates a detrimental effect on GB strength. The embrittling effect of H in presence of Mg in $\Sigma5$ (210) STGB is modified depending upon the

H site at the GB. However, at the opening free surface, H is expected to immediately diffuse to the position with the lowest segregation energy. This yields a higher embrittling energy compared to H remaining at the specific site of the GB (thin horizontal lines in the bars of Fig. 4c).

## Data availability

All data to evaluate the conclusions are present in the manuscript, the Extended Data items and the Supplementary Information. Raw data are available from the corresponding authors on reasonable request.

## Code availability

The code for this study can be found at ref. [46], which is also available on GitHub (https://github.com/oekosheri/GB_code).

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

**Acknowledgements** We acknowledge A. Sturm for technical support with cryo-experiments in the PFIB and cryo-suitcase transfer in the atom probe. The help of L. Stephenson for the cryo-transfer in the atom probe is also appreciated. We are grateful to D. Wan for the initial TDS measurements, and M. Adamek for the tensile experiments. B.G. acknowledges financial support from the ERC-CoG-SHINE-771602. Plane vector image in Fig. 1 was obtained from https://freesvg.org/plane-vector-image (public domain).

**Author contributions** H.Z., B.G., D.R. and D.P. developed the research concept; H.Z. was the lead experimental scientist of the study and interpreted the data; H.Z., B.G. and D.R. discussed and interpreted the APT results; P.C. and T.H. performed atomic calculations; B.S. conducted TDS measurements; C.-H.W. performed scanning Auger mapping measurements; H.Z., B.G., P.C. and T.H. wrote the manuscript. All authors contributed to the discussion of the results and commented on the manuscript.

**Funding** Open access funding provided by Max Planck Society.

**Competing interests** The authors declare no competing interests.

**Additional information**
**Correspondence and requests for materials** should be addressed to Huan Zhao, Baptiste Gault or Dierk Raabe.

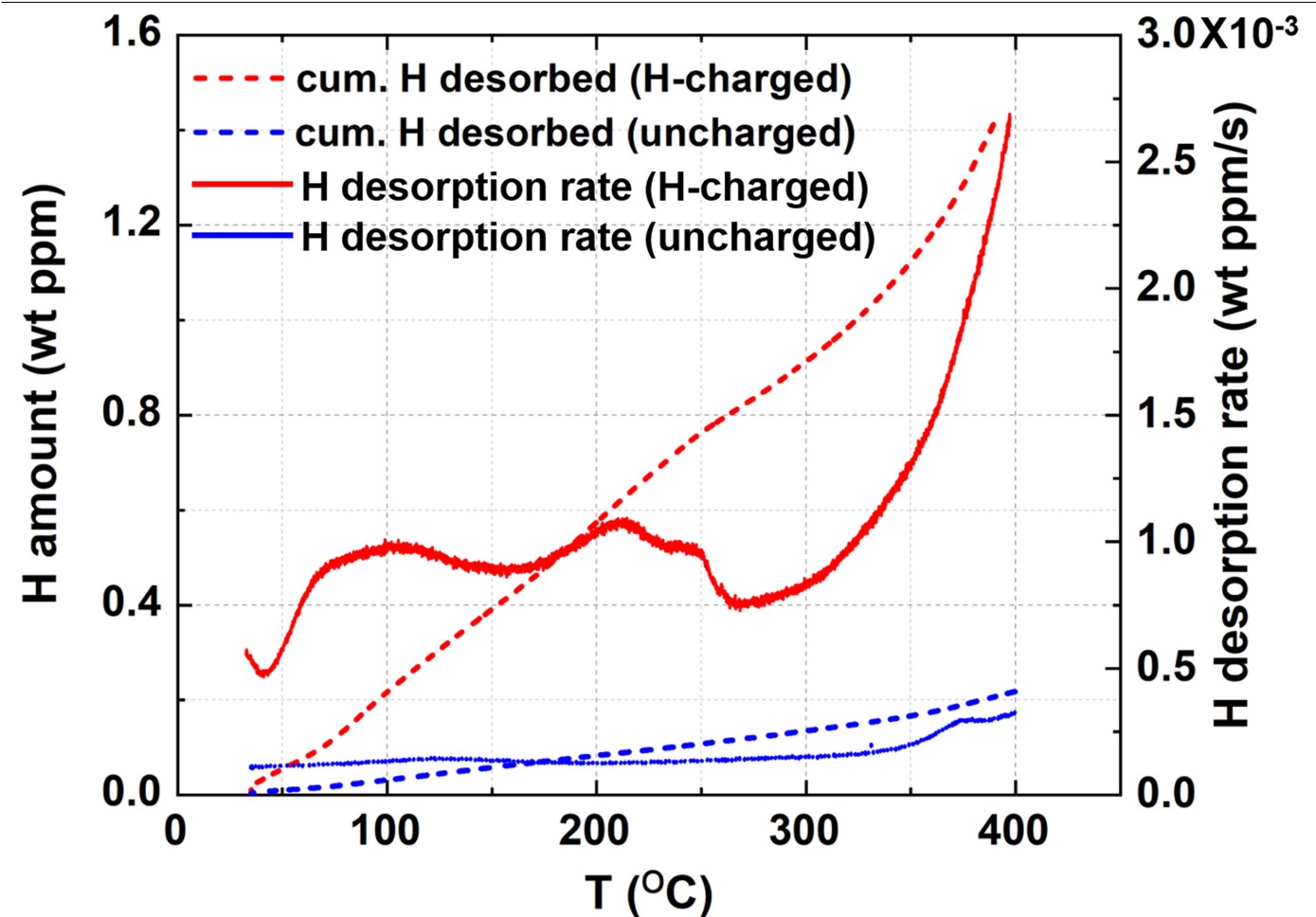

**Extended Data Fig. 1 | Thermal desorption spectroscopy analysis.** The H desorption spectra of uncharged and H-charged Al–Zn–Mg–Cu samples in the peak-aged state. cum., cumulative; ppm, parts per million.

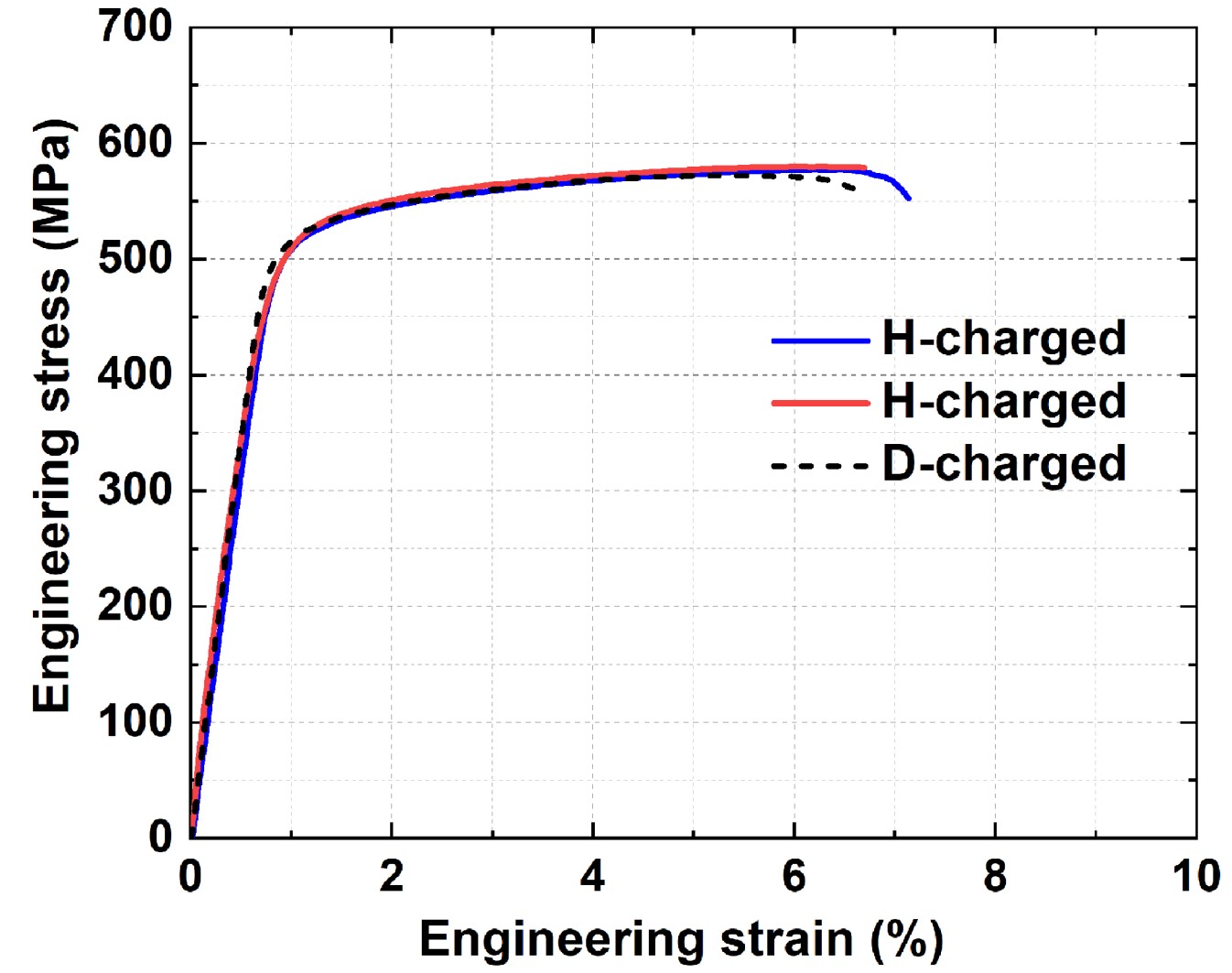

**Extended Data Fig. 2 | Tensile properties of H-charged and D-charged samples.** Engineering stress–strain curves of H-charged and D-charged Al–Zn–Mg–Cu samples in the peak-aged state showing that H and D have a similar embrittling effect on mechanical properties.

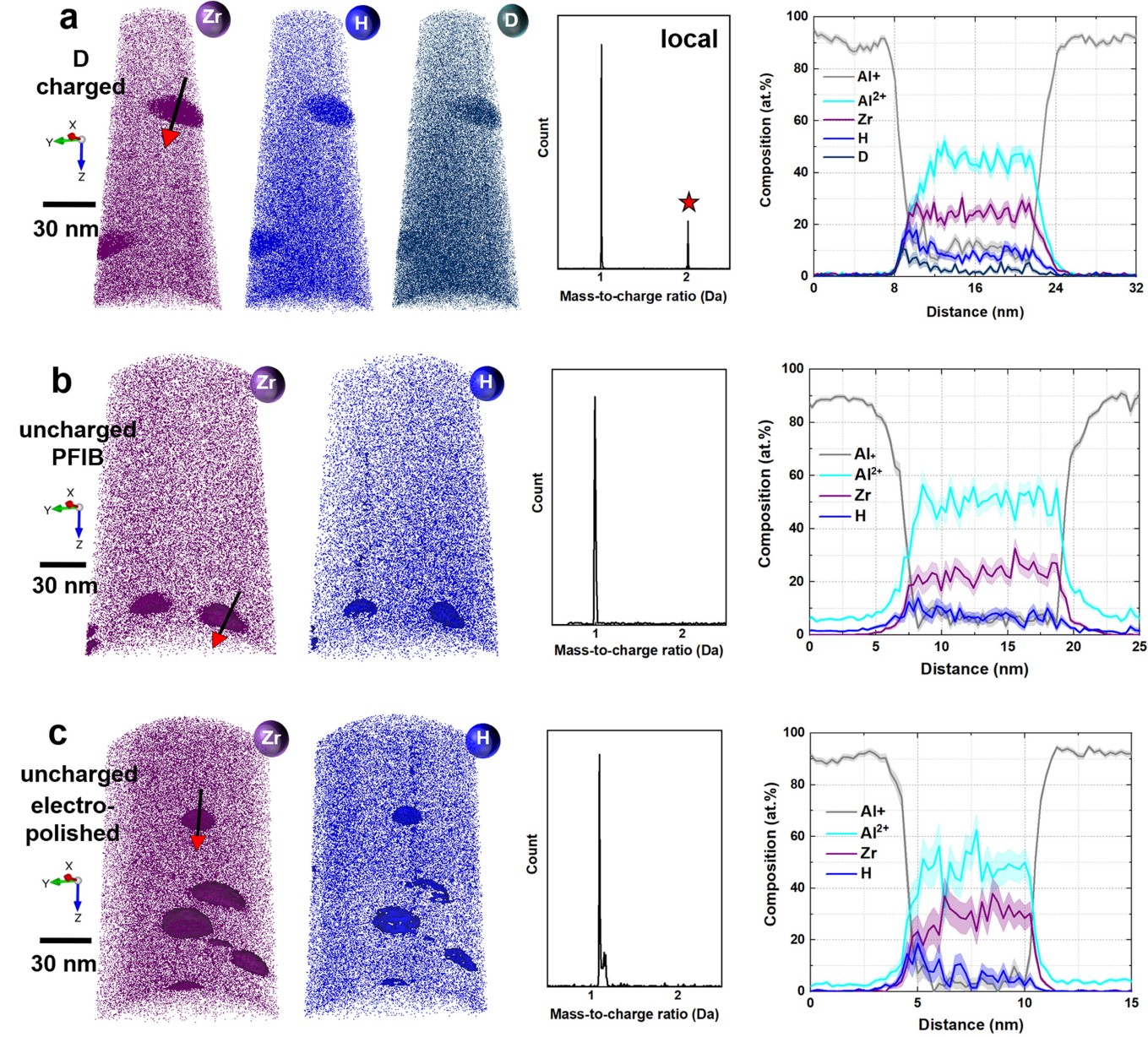

**Extended Data Fig. 3 | Atom probe analysis of Al₃Zr dispersoids in peak-aged Al–Zn–Mg–Cu samples. a**, D-charged. **b**, Uncharged sample prepared by PFIB. **c**, Uncharged samples prepared by electropolishing. Associated H peaks in the mass-to-charge ratio within the local Al₃Zr dispersoids (middle) and composition analysis across the Al₃Zr dispersoids (right) are also shown for each condition.

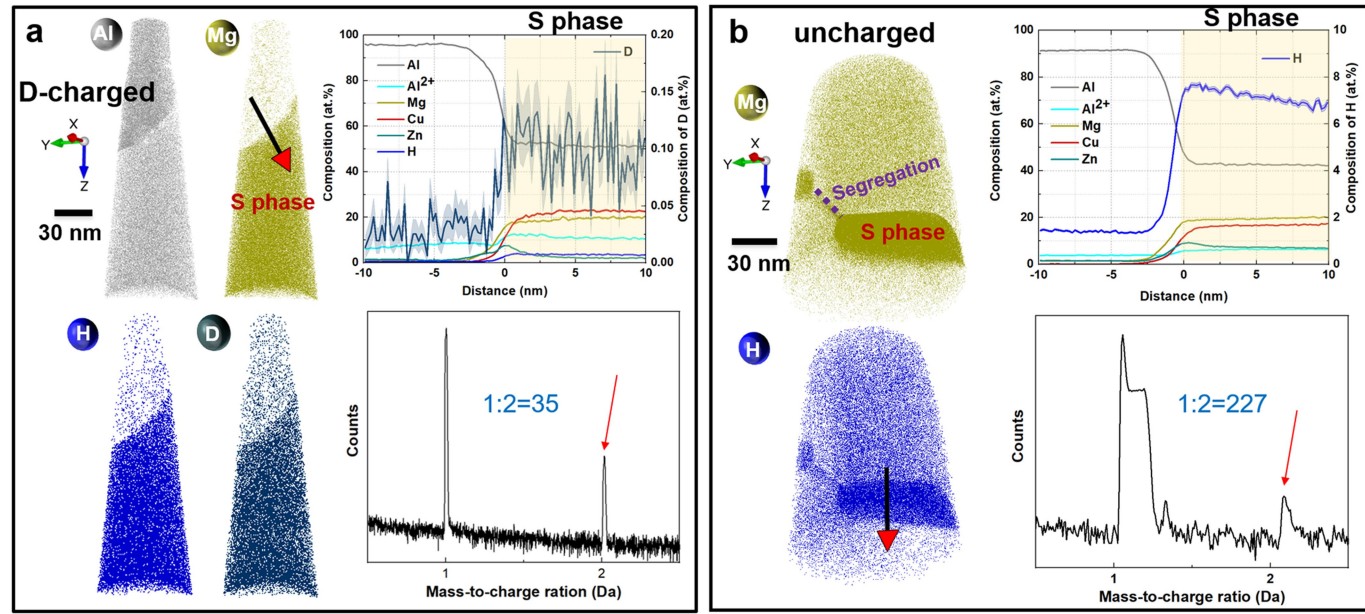

**Extended Data Fig. 4 | Atom probe analysis of S phase in Al–Zn–Mg–Cu samples. a**, D-charged; **b**, uncharged. Atom maps of Al, Mg, H and D are presented, with the S phase visualized by the Mg enriched regions. Associated H peaks in the mass-to-charge ratio within the S phases are also shown.

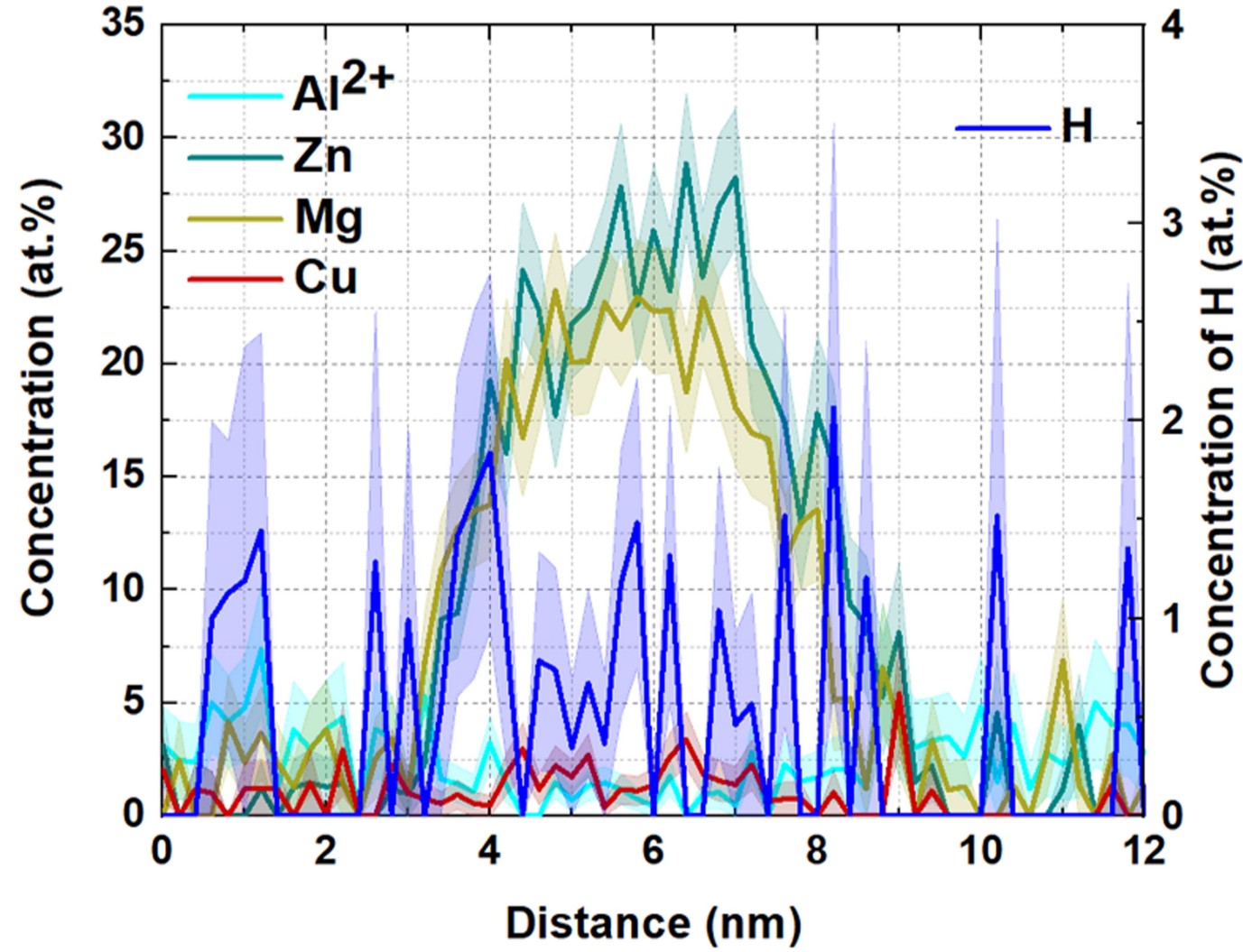

**Extended Data Fig. 5 | Atom probe analysis of bulk precipitates.** Representative composition profile across the bulk precipitate in D-charged Al–Zn–Mg–Cu samples in the peak-aged state, showing no H enriched.

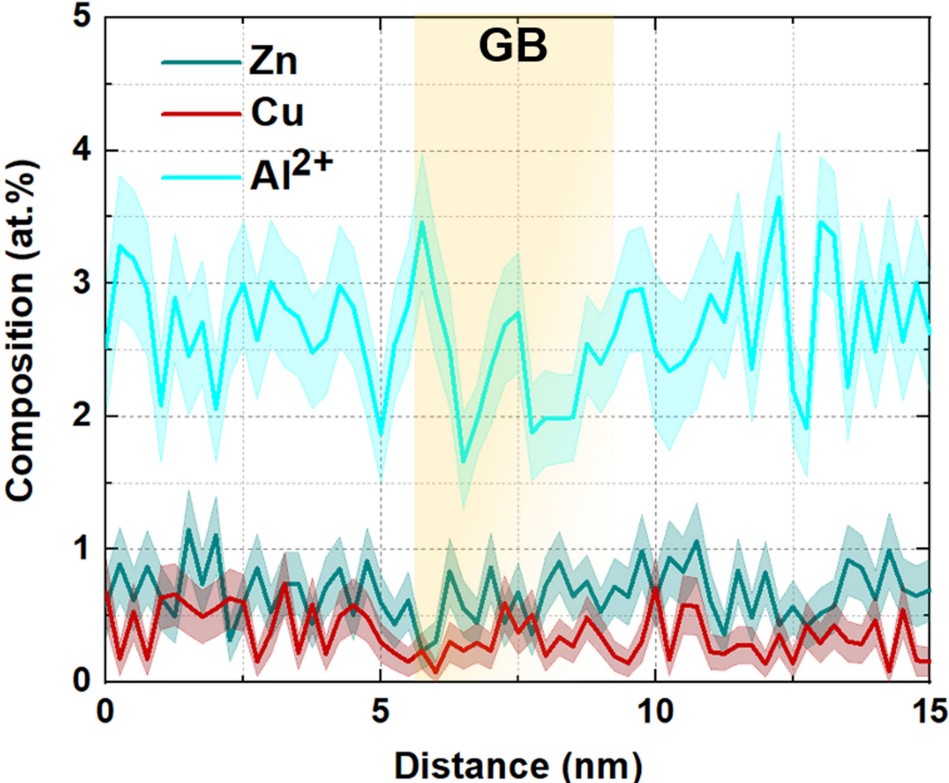

**Extended Data Fig. 6 | Atom probe analysis of the GB composition.** The mean chemical composition profile of Zn, Cu, $Al^{2+}$ across the GB represented in Fig. 3. The composition profile of $Al^{2+}$ shows the evaporation field not changing.

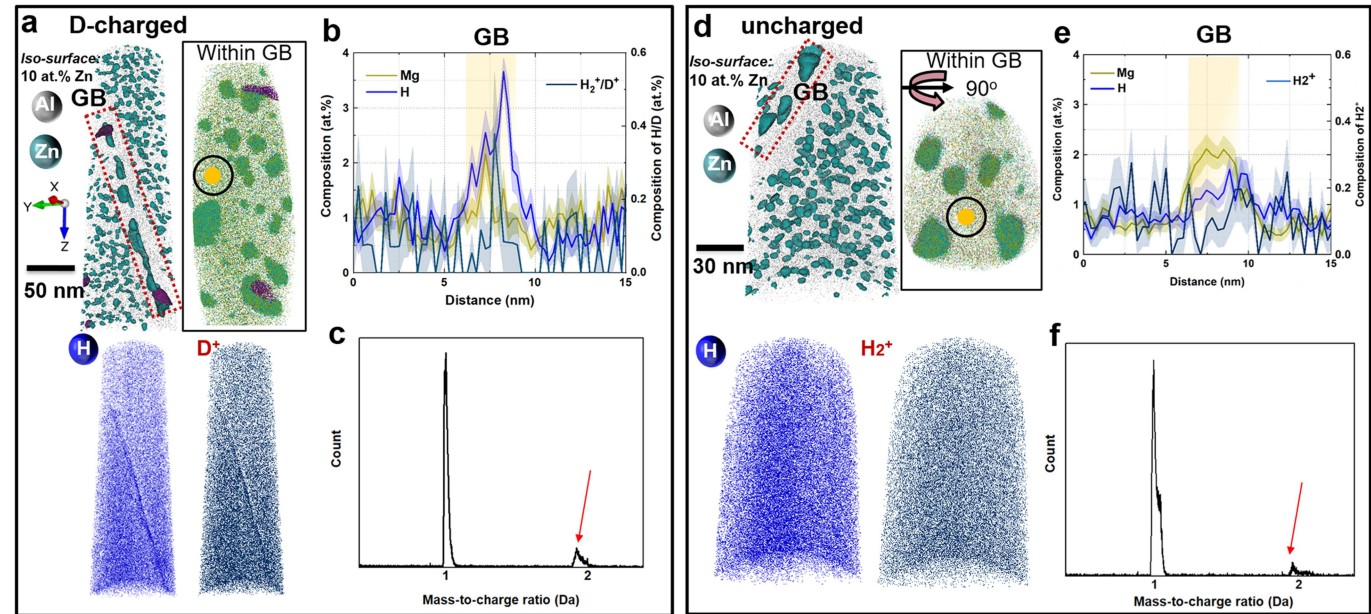

**Extended Data Fig. 7 | Atom probe analysis of GBs in peak-aged Al–Zn–Mg–Cu samples. a–c**, D-charged; **d–f**, uncharged. Associated H peaks in the mass-to-charge ratio within the GBs and composition analysis across the GB are also shown for each condition.

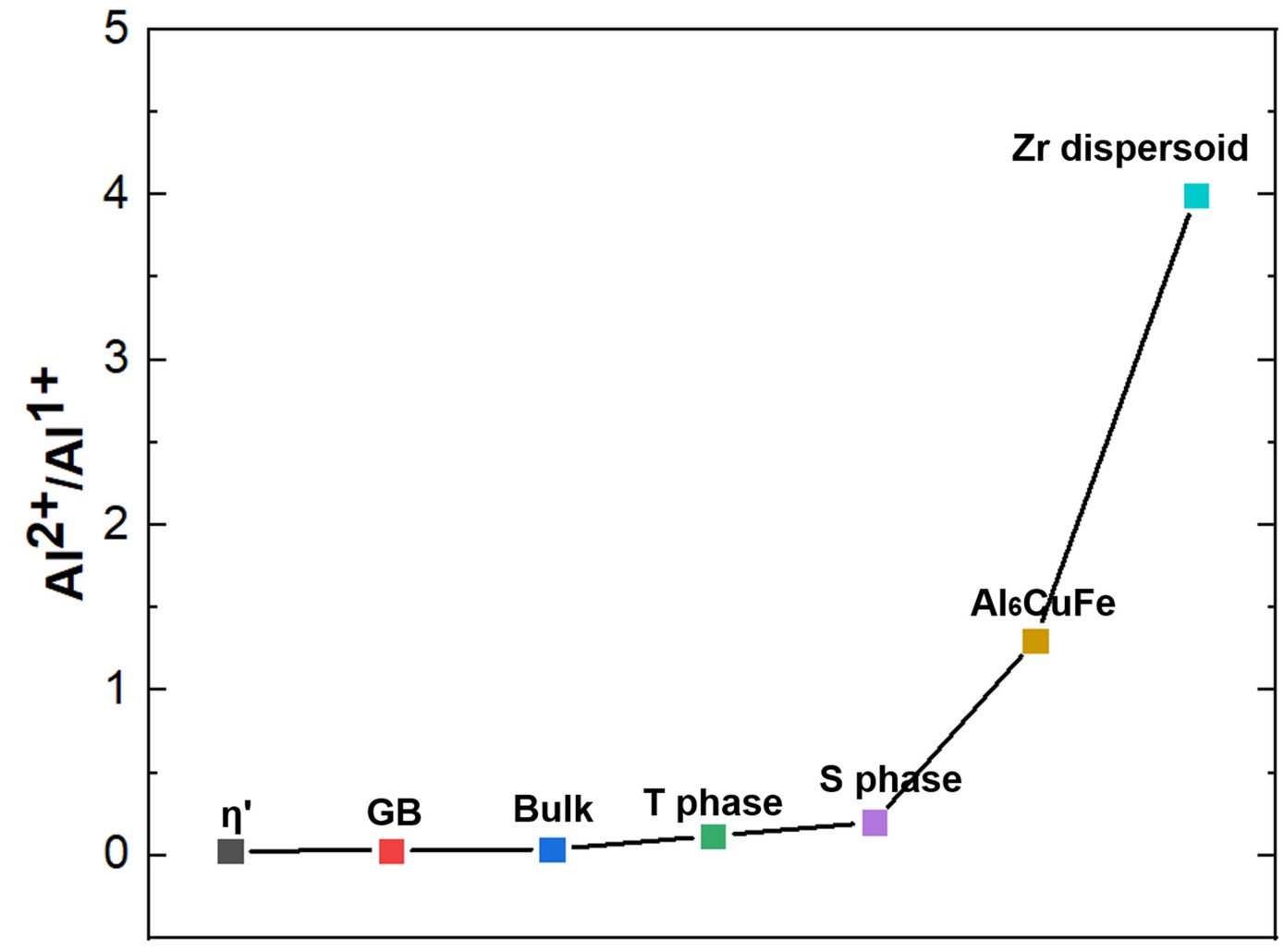

**Extended Data Fig. 8 | The local electrical field analysis for microstructural features.** The local electrical field for each microstructural feature was tracked by the charge state ratio of $Al^{2+}/Al^{+}$.

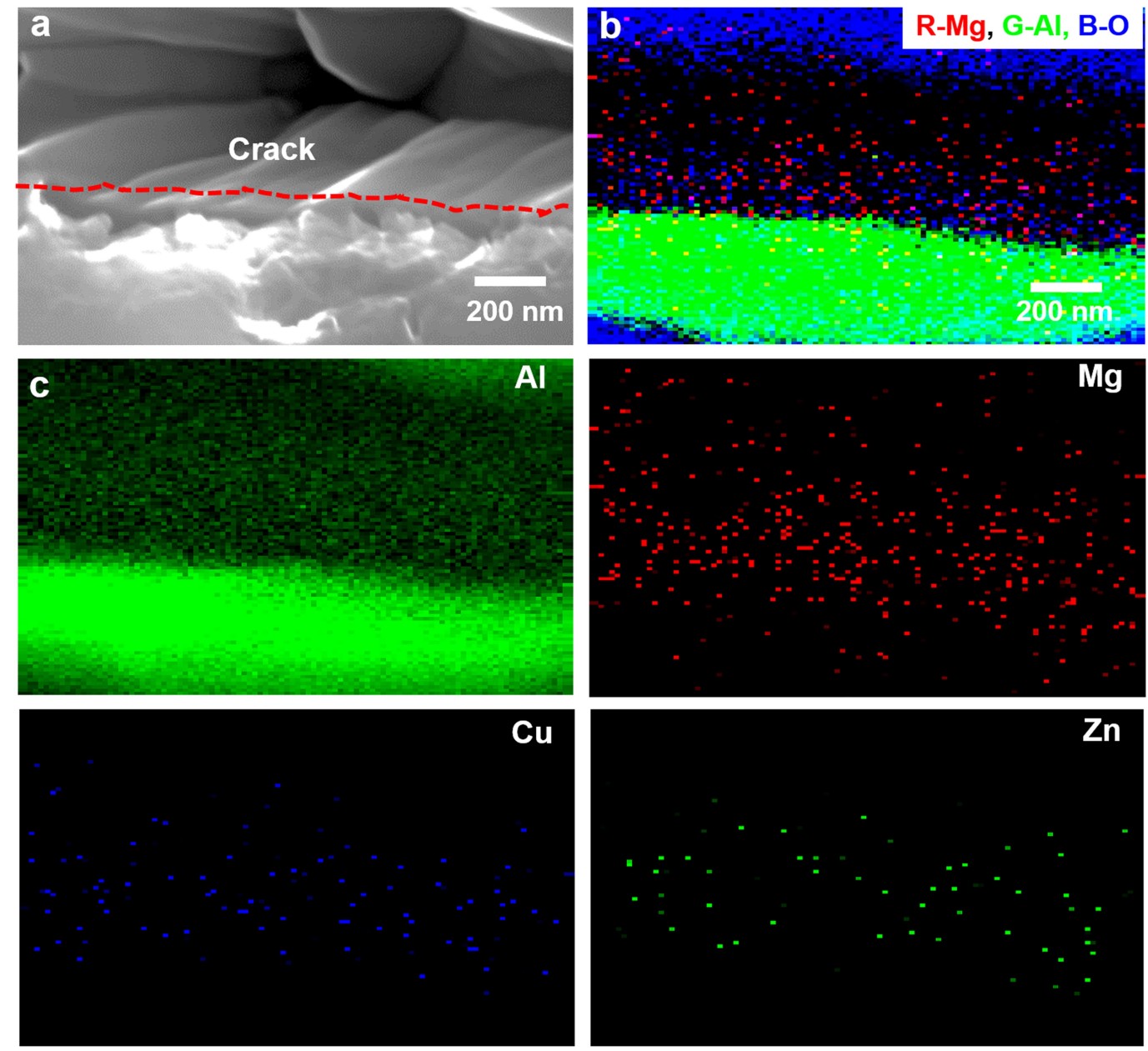

**Extended Data Fig. 9 | Scanning Auger electron microscopy analysis of a H-induced intergranular crack in the Al–Zn–Mg–Cu sample. a**, Scanning electron microscopy image of a crack at the grain boundary. **b**, Auger map of the overlay of Al, O and Mg showing Mg enriched at the crack. **c**, Elemental distribution images of Al, Zn, Mg, Cu at the crack.

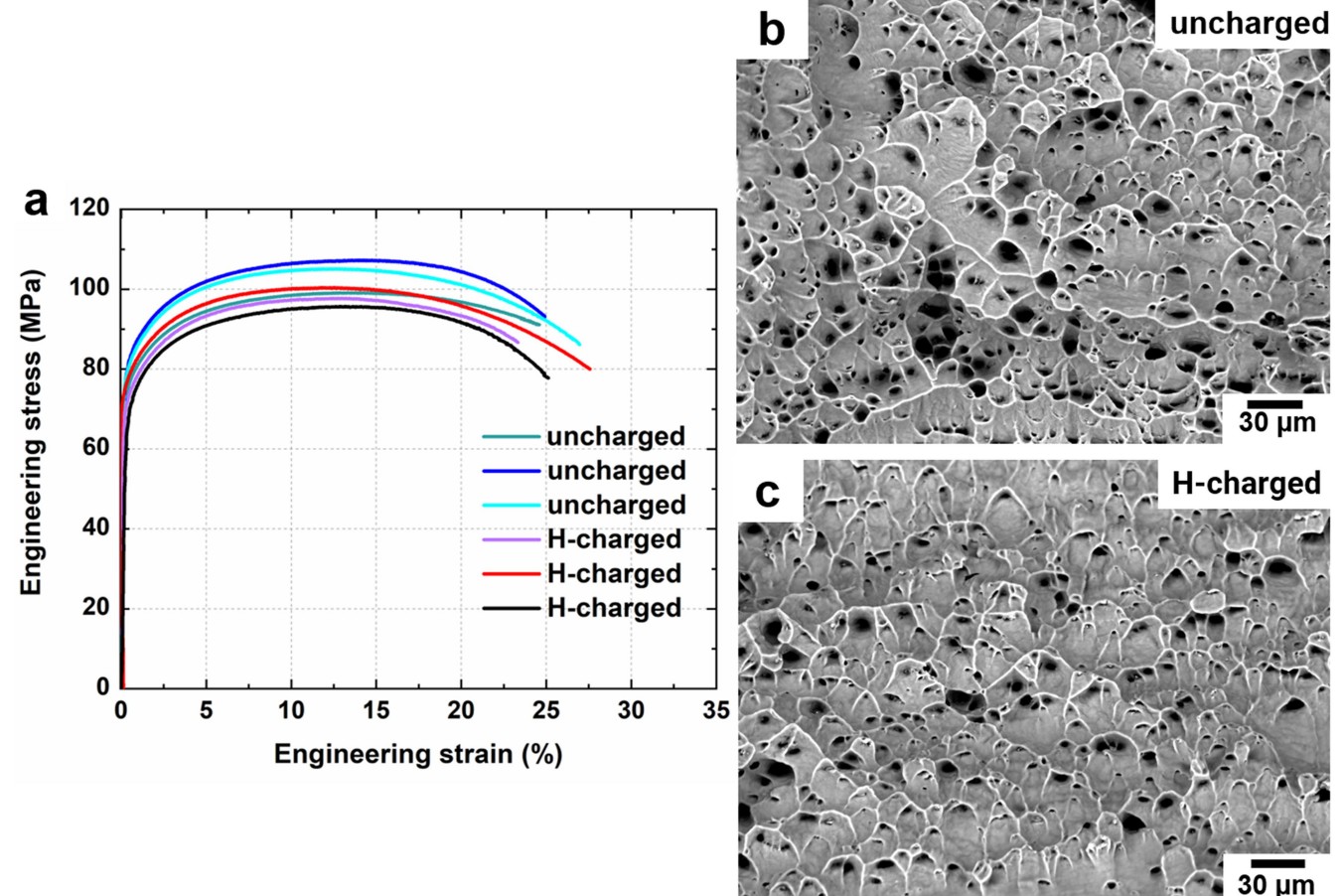

**Extended Data Fig. 10 | Tensile properties and metallographic fractography of Al–5.41 wt.% Zn alloy. a**, Engineering stress–strain curves of uncharged and H-charged Al–Zn samples. **b**, **c**, Typical scanning electron microscopy fractography of the uncharged (**b**), and H-charged (**c**) Al–Zn samples.