## [Peer Review File · Nature]

Manuscript Title: Hydrogen trapping and embrittlement in high-strength Al-alloys

Reviewer Comments & Author Rebuttals

Reviewer Reports on the Initial Version:

Referee #1:

Remarks to the Author:

This paper is concerned with the measurement of the location of H (and D) in a high strength 7xxx alloy using APT and the connection this has with the embrittling effect that H can have on this alloy. H can embrittle ferritic steels, Al alloys, Ni based superalloys, Zr alloys, etc and given its difficulty in measuring H using the common experimental techniques in Materials Science it is an important topic. Recent developments in cryo-APT have led to new measurements of H in steels and these works have appeared in Science (refs 18 and 19) and are now appearing in the field specific journals. This reviewer thinks that the present manuscript represents the first such H APT measurements in an Al alloys (but may be corrected by other reviewers) and this a novelty of the work. This is by no means a straightforward experimental measurement and hence is significant, but it must be acknowledged others have done this in other alloys such as steels which limits a bit the novelty (cryo-APT exists and has been used for this purpose in steels).

A potentially more significant contribution is the conclusions drawn that the embrittling effect of H is due to the co-location of Mg and H at the grain boundaries. In the view of this reviewer, this key point is not sufficiently well substantiated. If this is true, then it means alloys without Mg will not be susceptible to H embrittlement. This is easily tested by making an Al-Cu alloy with Si and Fe impurities so you still have the intermetallics, but no Mg to segregate to the grain boundaries. A fine precipitate hardened matrix can be made so this really important conclusion can be tested experimentally. Given the importance of this conclusion, this experiment should be done to substantiate this statement. This reviewer considers this conclusion to be potentially the most important contribution of this paper but requires further validation of the hypothesis.

In terms of the experiments, there are important questions that require clarification. The difference between the uncharged and charged tensile curves in Fig. S1 are not great. Typical error bars on the elongation of peak aged Al alloys is typically $\pm 2\%$ and when this is considered, the ductilities of the uncharged and charged alloys would overlap. So the embrittling effect of charging needs to be shown more convincingly to be statistically significant and multiple tensile curves should be shown etc.... The differences shown in Fig. S1 are much, much less than what is seen with high strength ferritic steels where the

embrittling effect is really catastrophic.

In Fig. 1, the failure is shown to be intergranular failure in the charged sample. The APT shows that even the uncharged material contains a lot of H. For example, the H found in the Zr dispersoids is basically the same whether you charge the samples or not. What is the failure mode in the uncharged sample? Is it also intergranular failure like Fig. 1a? Do you also find H at the grain boundary with Mg in the uncharged samples? Are their differences in the amounts compared to the charged samples? Given that the concentrations of D measured in the samples are very small and both charged and uncharged show high H contents, is the D really the difference causing the embrittling effect in Fig. S1, if these differences are statistically significant. This connection between the characterization and the cause of the potential difference in the tensile curves shown in Fig. S1 is not convincingly enough made.

There is an important lack of precision in the writing that should be addressed. Sometimes the authors talk of H-charging, other times D-charging. Sometimes uncharged and sometimes H-free (even though the uncharged contains lots of H). This confuses the story significantly and requires careful clarification. Also the first sentence should have been proof read before submitting to correct the English.

It is stated that Al alloys can reach the strengths of steels at half the weight. The absolute strongest Al alloys may have a yield strength around 600MPa. Steels can easily reach 1500MPa and some at 2000MPa. Al alloys can only reach the strengths of not-very-strong steels. What is the justification for saying 7xxx represent a near-optimal combination of strength-to-weight and formability? The forming is done in the STQ state and not when it is aged to have a good strength to weight. Many of these statements are imprecise and in some cases, incorrect.

In terms of motivation, 7xxx alloys are being considered for use in auto applications, particularly as the B-pillar. But the limiting factor is cost, not environmental embrittlement. The material currently used is high strength steels and high strength steels are more susceptible to H embrittlement than Al alloys. Environmental embrittlement is important and must be considered but it is an overstatement to say this is what is hindering widespread adoption.

Overall it is an interesting work showing the first APT measurements of H in Al alloys (to this reviewer's best knowledge) but important questions remain regarding the experiments, the statistical significance of the curves in Fig. S1, the failure mode of the uncharged sample and the key conclusion regarding the critical role of H and Mg must be substantiated by testing the hypothesis on an alloy without Mg.

Referee #2:

Remarks to the Author:

Summary of the key results

This paper provides identification of efficient hydrogen trapping sites on the basis of quantitative analyses:

- it clearly shows that intermetallic phases, Zr-rich dispersoids, and grain boundaries act as hydrogen traps;
- it gives relevant explanations on the trapping mechanisms.

Originality and significance: if not novel, please include reference

The paper is original and of great significance. It is of major interest in the current context.

Data & methodology: validity of approach, quality of data, quality of presentation

There is no problem concerning the data and methodology. However, the presentation of the manuscript should be improved. The authors should distinguish different sections: abstract, introduction, experimental, results and discussion, conclusions.

Appropriate use of statistics and treatment of uncertainties

The authors should address, in the body of the manuscript, the representativeness of their analyses. This is a major issue. The technique used is really interesting, but the volume analyzed is reduced. Comments on that point are required in the body of the manuscript.

Conclusions: robustness, validity, reliability

Globally, the robustness of the work is clear. A short discussion about the representativeness of the analyses is mandatory. The discussion concerning the segregation energy / embrittling energy and effect of Mg is not clear. This section needs to be improved. Maybe the reviewer has not understood the text, but there is something unclear in "Upon integration of the H concentrations over all possible sites in the unit area, a total embrittling energy of $\sim 0.6 \text{ J/m}^2$ for this GB is obtained. This value is substantially more positive (i.e., more detrimental) than the embrittling energy determined for the Zr dispersoids ($\sim 2 \text{ J/m}^2$). It is also larger than for the S phase ($\sim 129 \text{ mJ/m}^2$),..."

Suggested improvements: experiments, data for possible revision

See previous comments:

- organization of the text
- discussion about the representativeness of the results
- section focused on embrittling/segregation energies with and without Mg

References: appropriate credit to previous work

The authors forgot to cite relevant references concerning H trapping in aluminum alloys, e.g. AA2024.

Clarity and context: lucidity of abstract/summary, appropriateness of abstract, introduction and conclusions

The context is clearly described. The conclusions and suggestions for future work are very interesting. However, the conclusions concerning H ingress should be changed, i.e. "Al-alloys can absorb H during processing....leads to H ingress": the authors evidenced the presence of residual H in the Al alloys, but did not study the mechanisms. The comment concerning the influence of temperature is speculative, based on reference to literature, but this has not been studied in the paper.

Referee #3:

Remarks to the Author:

The authors provide atom probe tomography (APT) measurements of trapping sites in a 7xxx-Al high-strength alloy, in an attempt to establish a mechanistic understanding of H-embrittlement in such alloys. This has been particularly a challenging topic for the community due to the low solubility of H in this alloy as well as the experimental difficulty in achieving spatially-resolved characterization of H at specific microstructural features. The authors also couple the experimental work with DFT calculations to explain the enrichment of H observed in intermetallic phases as well as compute the embrittling energy associated with H.

The work is of high quality and generally speaking the paper can be easily understood by experts in the field. The general notion of H enrichment at defects and grain boundaries is not necessary new from a theoretical point of view and it has been well accepted. But the work presented clearly provides direct evidence of this. Perhaps the real major conclude of this work is that insights the experiments and the simulations suggest that "when combined with Mg, the strong driving force for H to segregate to a free surface with respect to a possible interstitial site at GBs in presence of Mg, favors GB decohesion and drives the system towards crack formation." However, I question whether this work is general enough for the general readership of Nature.

Additional points to be addressed:

1) Figure 1C, the abbreviation PFZs was never defined. Also not clear how the PFZs are of relevant to the current study so what is the emphasis of Fig 1C?

2) The authors discuss that H has low solubility in Al. However the focus of the paper is on H enrichment at bulk features. No discussion is made to explain to the reader how H would diffuse to these features at all with the exception of high-temperature processing towards the end of the manuscript. However, how H would ingress in samples present in "aqueous or humid environments" has never been discussed given the low solubility of H at room temperature

3) The discussion about the comparison between the uncharged specimens and charged specimens is not really clear. Looking at Fig S2 it seems that the H content in Zr-dispersoids is the same for the charged and uncharged samples. So what is the conclusion here? The same is for charged and uncharged S phase. I think this discussion needs to be fully revised to make it clear to the reader what is different in both cases! Additionally, no discussion or results on comparison between H content in grain boundaries of charged and uncharged samples. It seems that this would be an important case to show/discuss. After all the main conclusion is about grain boundary embrittlement. Maybe having a table that summarizes the comparisons for the H content in each defect in the charged and uncharged states would also help the reader.

4) Minor point: No discussion about fig 3D?

5) In the conclusion the authors state "The results shed light on possible approaches for the further control and design of precipitations and GB segregation to render an improved resistance to H-induced damage." ... This seems like a general open-ended statement. Point out to the reader what are such possible approaches based on the current results.

Author Rebuttals to Initial Comments:

Referee #1 (Remarks to the Author):

Opening comment: This paper is concerned with the measurement of the location of H (and D) in a high strength 7xxx alloy using APT and the connection this has with the embrittling effect that H can have on this alloy. H can embrittle ferritic steels, Al alloys, Ni based superalloys, Zr alloys, etc and given its difficulty in measuring H using the common experimental techniques in Materials Science it is an important topic. Recent developments in cryo-APT have led to new measurements of H in steels and these works have appeared in Science (refs 18 and 19) and are now appearing in the field specific journals. This reviewer thinks that the present manuscript represents the first such H APT measurements in an Al alloys (but may be corrected by other reviewers) and this a novelty of the work. This is by no means a straightforward experimental measurement and hence is significant, but it must be acknowledged others have done this in other alloys such as steels which limits a bit the novelty (cryo-APT exists and has been used for this purpose in steels).

We are most grateful for these very positive and supportive comments from the reviewer.

We highly appreciate that the reviewer shares our view that the manuscript represents the

first such H APT measurements in Al alloys, establishing a high level of novelty. We also thank the reviewer for acknowledging the extreme difficulty of the experiments, namely, measuring H in Al alloys, and he/she also emphasizes the broader significance of this work.

We fully acknowledge that cryo-APT has been used before for the detection of H in steels¹⁻³, including works by our group. APT had previously been used for the study of H in steel and multilayers without using cryo-enabled workflows^{4,5}. However, we would like to differentiate this study from previous studies in steels using cryo-APT. H embrittlement in steels has been extensively studied and their embrittlement behavior is overall better documented. The studies reported in Ref. 1–3 however lacked correlation with respect to the changes in the engineering behavior associated with H and direct experiment-guided theoretical insights.

Unlike for steel, H embrittlement studies for Al alloys are very scarce. Knowledge about H-microstructure interactions and the impact of H on the embrittlement behavior of Al-alloys are thus missing. The key question about H embrittlement remained open so far for the case of Al-alloys, namely, whether H poses any threat at all for this material class, and if so in which Al alloy series it has an embrittling effect. These are the questions we can now answer with this study, by not only proving the first direct evidence of H distribution inside a complex engineering Al alloy but also through corresponding atomistic simulations which reveal the role and importance of H-traps inside of the material. This combined theoretical-experimental approach paves the way for designing better H-resistant high-strength Al alloys by adjusting the phase fractions and distributions guided by these findings.

Comment 1: A potentially more significant contribution is the conclusions drawn that the embrittling effect of H is due to the co-location of Mg and H at the grain boundaries. In the view of this reviewer, this key point is not sufficiently well substantiated. If this is true, then it means alloys without Mg will not be susceptible to H embrittlement. This is easily tested by making an Al-Cu alloy with Si and Fe impurities so you still have the intermetallics, but no Mg to segregate to the grain boundaries. A fine precipitate hardened matrix can be made so this really important conclusion can be tested experimentally. Given the importance of this conclusion, this experiment should be done to substantiate this statement. This reviewer considers this conclusion to be potentially the most important contribution of this paper but requires further validation of the hypothesis.

Answer: We highly appreciate that the reviewer acknowledges the significant contribution of this work to understanding the embrittlement effect at grain boundaries (GBs). We thank the reviewer for suggesting the experiment of testing Al-Cu alloys (with Si and Fe impurities). However, we think the Al-Cu alloy is a quite different system, and we do not suggest in our manuscript that H embrittlement will not occur in Al-alloys without Mg. Note that the microstructures, chemical composition, distribution of precipitates, defects, and heat treatments etc. are also all potentially important factors in determining the H embrittlement behavior of materials. The Al-Cu alloy system has the precipitation sequence of GP- θ' - θ (Al_2Cu). Each of these precipitates is plate-shaped and exhibits a large aspect ratio ⁶. *The morphology is often rationalized by a large anisotropy in the interfacial energy and, elastic strain energy, making these precipitates likely traps for H.* Previous studies about Al-Cu alloys showed that dislocations, GBs and stable θ precipitates can act as short-circuit diffusion pathways and trapping sites for H ⁷⁻¹¹.

Indeed, although we do not quite agree that the specific Al-Cu system suggested by the referee is an ideal model system in the current context, performing additional experiments is an excellent idea and we have fully complied and done so. We employed scanning Auger electron microscopy to investigate the elemental distribution of the intergranular crack of our H-charged Al-Zn-Mg-Cu alloy after tensile fracture, which provides high chemical sensitivity combined with a spatial resolution in the nanometer-range. The results can be seen below and also in Fig. S13 in the revised manuscript. Fig. R1(A) shows the scanning electron micrograph of the H-induced intergranular crack. The Auger mapping performed at the crack in Fig. R1(B) shows a clear difference in the spatial distribution at the crack and inside the matrix. The chemical mapping by the Auger probe provides direct visualization of enrichment of Mg at the crack (Mg/Al peak ratio of 4.7%), which is even stronger than it at GBs (Mg/Al concentration ratio of 2.2%, Fig. 3F). This supports our interpretation of the joint embrittling effect of the Mg and H at GBs. The concentration of Mg is much higher than that of Zn and Cu, while O shows obvious enrichment at the crack surface, as expected.

Fig. R1. Scanning Auger electron microscopy analysis of a H-induced intergranular crack in the Al-Zn-Mg-Cu sample (A) Scanning electron microscopy image of a crack at the grain boundary; (B) Auger map of the overlay of Al (green), O (blue) and Mg (Red) showing Mg enriched at the crack; (C) Elemental distribution images of Al, Zn, Mg, Cu at the crack.

We added this item as additional information to Page 10 in the modified manuscript:

“Further investigation on the elemental distribution at a H-induced intergranular crack using scanning Auger electron microscopy reveals the enrichment of Mg at the cracked GB. The enrichment is even stronger (by a factor of 2) than the Mg concentration at the GB (Fig. 3F), which supports the finding that the co-segregation of Mg and H to free surfaces provides the driving force for the embrittlement of GBs (Fig. S13).”

Comment 2: In terms of the experiments, there are important questions that require clarification. The difference between the uncharged and charged tensile curves in Fig. S1 are not great. Typical error bars on the elongation of peak aged Al alloys is typically +-2% and when this is considered, the ductilities of the uncharged and charged alloys would overlap. So the embrittling effect of charging needs to be shown more convincingly to be statistically significant and multiple tensile curves should be shown etc.... The differences shown in Fig. S1 are much, much less than what is seen with high strength ferritic steels where the embrittling effect is really catastrophic.

Answer: We thank the reviewer for raising the point about statistics.

We fully comply and conducted more tensile experiments on the H-charged and uncharged samples, which are shown below and in Fig. 1(A) in the revised manuscript. The ductility of the uncharged and H-charged samples is around 13.6% and 6.6%, respectively. We compared the reduction of elongation as defined by the equation below. A 51.5% elongation loss is shown in the H-charged samples as compared with the uncharged samples. This is a significant embrittling effect. Note that the diffusivity of H in Al is much lower than that in ferritic steels. H cannot be saturated with the current charging setup even if we prepare a very thin sample. Therefore, only the surface area is affected by H in Al, thus the overall tensile property does not seem to degrade as much as in ferritic steels.

$$\text{Embrittlement index } I_e: I_e = \frac{A_{\text{no H}} - A_{\text{with H}}}{A_{\text{no H}}} \times 100\% \quad [1]$$

A: total elongation

Fig. R2. Engineering stress-strain curves of uncharged and H-charged Al-Zn-Mg-Cu samples in the peak aged state.

Comment 3: In Fig. 1, the failure is shown to be intergranular failure in the charged sample. The APT shows that even the uncharged material contains a lot of H. For example, the H found in the Zr dispersoids is basically the same whether you charge the samples or not. What is the failure mode in the uncharged sample? Is it also intergranular failure like Fig. 1a? Do you also find H at the grain boundary with Mg in the uncharged samples? Are their differences in the amounts compared to the charged samples? Given that the concentrations of D measured in the samples are very small and both charged and uncharged show high H contents, is the D really the difference causing the embrittling effect in Fig. S1, if these differences are statistically significant. This connection between the characterization and the cause of the potential difference in the tensile curves shown in Fig. S1 is not convincingly enough made.

Answer: We thank the reviewer for the question about the fracture mode of the uncharged samples. A typical scanning electron micrograph of the fracture surface is shown below. The uncharged samples show a transgranular ductile, dimple-type fracture surface (Fig. R3A). Near this fracture zone, microvoids and/or blunted ductile microcracks (Fig. R3B) mainly occur at plastic shear regions (Fig. R3C), intermetallic particles, and GBs. Note that some portion of the observed strain-induced crack can be attributed to the resulting high stress/strain

concentrations near the slip planes ¹². In contrast, a predominantly intergranular fracture is observed in the fracture surface of the H-charged specimen (Fig. R3D). The size of the intergranular facets is similar to the grain size. The intergranular fracture behavior is driven by the crack formation at GBs (Fig. R3E-F).

Fig. R3. (A-B) Typical scanning electron microscopy fractography of the uncharged sample; (C) Electron backscatter diffraction imaging of the enlarged rectangle region in (B) showing the transgranular crack along slip planes; (D) Typical scanning electron microscopy fractography of the H-charged sample; (E-F) Backscattered electron imaging and electron backscatter diffraction imaging of the intergranular crack.

We used D to avoid several potential H artifacts in the atom probe analysis, e.g. the introduction of spurious H from the sample preparation and the residual gas in the atom probe chamber after validating that H and D show the similar embrittling effect in mechanical properties (Fig. R4). We analyzed D-charged samples and uncharged reference samples, and confirmed the presence of D at a mass-to-charge ratio of 2 Da by comparing the mass spectra.

The reviewer is correct that H is found in the Zr-rich dispersoids, with a similar amount in the D-charged and uncharged samples. However, only a peak of the mass-to-charge ratio at 1 Da is detected in the dispersoids in uncharged specimens. In the D-charged material, a distinct peak at 2 Da appears in the Zr-rich dispersoids, which gives evidence of the presence of D (Fig. S3). The presence of H in these particles is related to the high affinity of Zr for H, and the high solubility of H in these dispersoids. The H comes likely from the processing or the specimen preparation – see the discussion in Breen et al. ³, where we systematically investigated all possible influencing factors and pathways for H uptake during such experiments.

Fig. R4. Engineering stress-strain curves of H-charged and D-charged Al-Zn-Mg-Cu samples in the peak aged state.

The comparison of the H and D distribution at GBs between the uncharged and D-charged specimens is shown below and in Fig. S9 in the modified manuscript. For the uncharged sample, we also observe H in the mass spectrum. Fig. R5E shows that a slight H enrichment is found at the GB in the uncharged samples, while no clear indication of the enrichment of the

peak at 2 Da is observed. Again, these could either be coming from H trapped inside the material itself or from the specimen preparation route. A higher signal ratio at the peak at 2 Da (by a factor of 3) is observed in the D-charged sample and supports that D is indeed enriched at the GB. We obtained 7 such APT datasets taken at GBs in D-charged samples, and all show consistent enrichment of H and D at GBs (2 more shown in Fig. S10,11).

We thank the reviewer for raising the question on the connection of the measured D content with the tensile properties. We could see the same 51.5% elongation reduction of the D-charged samples compared to the uncharged ones as shown above (Fig.R4).

Fig. R5. APT analysis of GBs in peak aged Al-Zn-Mg-Cu samples (A-C) D-charged; (D-F) uncharged. Associating H peaks in the mass-to-charge ratio within the GBs and composition analysis across the GB are also shown for each condition.

Comment 4: There is an important lack of precision in the writing that should be addressed. Sometimes the authors talk of H-charging, other times D-charging. Sometimes uncharged and sometimes H-free (even though the uncharged contains lots of H). This confuses the story significantly and requires careful clarification. Also the first sentence should have been proof read before submitting to correct the English. It is stated that Al alloys can reach the strengths of steels at half the weight. The absolute strongest Al alloys may have a yield strength around 600MPa. Steels can easily reach 1500MPa and some at 2000MPa. Al alloys can only reach the strengths of not-very-strong steels. What is the justification for saying 7xxx represent a near-optimal combination of strength-to-weight and formability? The forming is done in the STQ

state and not when it is aged to have a good strength to weight. Many of these statements are imprecise and in some cases, incorrect.

Answer: We cordially thank the reviewer for the helpful comments. We fully comply. The manuscript was carefully checked and rewritten. We removed the sentence of *"If pure aluminium is soft, when alloyed with small amounts of carefully selected elements, it can reach similar strength to e.g. steels at only half the part's weight."* and *"High strength Al-alloys from the 7xxx series represent a near-optimal combination of high strength to weight ratio"*.

We also fully comply on the suggestion for terminology and consistently use H-charged, D-charged and uncharged for the different conditions in the modified manuscript. The tensile tests were performed on the H-charged specimens to show that H has an embrittlement effect on this alloy. Detailed atom probe analysis was carried out on the D-charged samples to avoid several potential types of H artifacts, e.g. the spurious introduction of H from the sample preparation and the residual gas in the atom probe after validating that H and D show the similar embrittling effect in mechanical properties (Fig. S2). We make these points clearer in the revised manuscript.

Page 2: "Here, we address these critical questions by using the latest developments in cryo-atom probe tomography (APT), enabled by cryo-plasma focused-ion beam (PFIB) specimen preparation to investigate H associated with different microstructures in an Al-alloy. Through isotope-labelling with deuterium (D), we partly avoid characterization artifacts associated to the introduction of H from the sample preparation^{3,13} and from residual gas in the atom probe vacuum chamber."

Page 3: "We studied a 7xxx Al-alloy with a composition of Al-6.22Zn-2.46Mg-2.13Cu-0.155Zr (wt.%) in its peak-aged condition. For this alloy, electrochemical charging with H leads to a critical drop in the ductility compared with uncharged samples (Fig. 1A)."

Page 4: "Peak-aged specimens were electrochemically charged with D for subsequent APT probing after validating that H and D show a similar embrittling effect on the mechanical properties (Fig. S2). D-charged specimens were prepared by PFIB, to avoid contamination by

gallium, at cryogenic temperatures to limit the introduction of H¹³, and immediately analyzed by APT using voltage pulsing to minimize residual H from APT^{3,13}.”

Comment 5: In terms of motivation, 7xxx alloys are being considered for use in auto applications, particularly as the B-pillar. But the limiting factor is cost, not environmental embrittlement. The material currently used is high strength steels and high strength steels are more susceptible to H embrittlement than Al alloys. Environmental embrittlement is important and must be considered but it is an overstatement to say this is what is hindering widespread adoption.

Answer: We thank the reviewer for this comment, and have modified the manuscript.

Page 1: *“Ever more stringent regulations on greenhouse gas emissions from transportation impose to revisit materials constituting vehicles¹⁴. High-strength Al-alloys, often used in aircrafts, could help reduce the weight of automobiles, but are susceptible to environmental degradation.”*

Page 2: *“High-strength Al-alloys of the 7xxx-series are essential structural materials in aerospace, manufacturing, transportation and mobile communication^{15,16}, due to their high strength-to-weight ratio, which enables products with lower fuel consumption and environmental impact.”*

Overall it is an interesting work showing the first APT measurements of H in Al alloys (to this reviewer's best knowledge) but important questions remain regarding the experiments, the statistical significance of the curves in Fig. S1, the failure mode of the uncharged sample and the key conclusion regarding the critical role of H and Mg must be substantiated by testing the hypothesis on an alloy without Mg.

Answer: We cordially thank the reviewer for the very positive comments, the careful analysis and most pertinent and helpful recommendations which greatly helped us to improve the manuscript. We addressed these issues regarding the tensile curves, the failure mode of the uncharged samples and used scanning Auger microscopy at the crack region to provide

additional proof for the effect of Mg, as presented above. We hope that all the reviewer's concerns have been properly addressed and we will be ready to make further efforts to improve the quality of the manuscript, should further points arise.

Referee #2 (Remarks to the Author):

- Summary of the key results

This paper provides identification of efficient hydrogen trapping sites on the basis of quantitative analyses:

- it clearly shows that intermetallic phases, Zr-rich dispersoids, and grain boundaries act as hydrogen traps;
- it gives relevant explanations on the trapping mechanisms.

- Originality and significance: if not novel, please include reference

The paper is original and of great significance. It is of major interest in the current context.

We very cordially thank the reviewer for the strong support and for the important comments.

- Data & methodology: validity of approach, quality of data, quality of presentation

There is no problem concerning the data and methodology. However, the presentation of the manuscript should be improved. The authors should distinguish different sections: abstract, introduction, experimental, results and discussion, conclusions.

Answer: We thank the review for this helpful comment. Subtitles on the different sections have been added to the modified manuscript.

- Appropriate use of statistics and treatment of uncertainties

The authors should address, in the body of the manuscript, the representativeness of their analyses. This is a major issue. The technique used is really interesting, but the volume analyzed is reduced. Comments on that point are required in the body of the manuscript.

Answer: We would like to thank the reviewer for the comment. We fully comply and add more data to the supplemental part of the paper. Our other datasets show that the data presented in the bulk manuscript are representative of our atom probe analysis. For the H distribution at grain boundaries (GBs), we have altogether obtained 7 full high-quality APT datasets from D-charged samples, with two more shown below. These observations are all consistent with the analysis shown in Fig. 3D in the main manuscript. i.e. the H concentration in the bulk matrix remains low and the fine and coarse (Mg,Zn)-rich strengthening precipitates do not show relevant H enrichment. The solute content analysis across the GB at the region between precipitates in Fig. S10,11(C) shows that the GBs are enriched with Mg in the range of 2.5-3 at.% Mg. D(H₂⁺) atoms are also shown enriched at the GB region. For the enrichment of H at the intermetallic phases, more datasets have been shown in Fig. S3-6 in the supplemental material.

Fig. S10. APT analysis of a GB in the peak aged Al-Zn-Mg-Cu sample subjected to D-charging (120°C/24h). (A) A set of isosurfaces highlights the dispersion of fine Zn-rich precipitates in the matrix and of the larger ones at the GB. (B) Atom maps of Mg, H and D(H₂⁺). (C) Solute distribution at the GB plane, (D) Representative solute composition profile across the GB in between of precipitates shown in the region highlighted by the circle.

Fig. S11. APT analysis of a GB in the peak aged Al-Zn-Mg-Cu sample subjected to D-charging (120°C/24h). (A) A set of isosurfaces highlighting the dispersion of fine Zn-rich precipitates in the matrix and the larger ones at the GB. (B) Atom maps of Mg, H and D(H₂⁺). (C) Solute distribution at the GB plane, (D) Representative solute composition profile across the GB in between of precipitates shown in the region emphasized by the circle.

We added these two datasets and the corresponding discussion items to the modified manuscript (Fig. S10,11).

Page 6: "We obtained 7 APT datasets containing GBs in D-charged samples, and all show consistent enrichment of H and D at GBs (2 additional datasets are shown in Fig. S10,11)."

- **Conclusions: robustness, validity, reliability**

-Globally, the robustness of the work is clear. A short discussion about the representativeness of the analyses is mandatory. The discussion concerning the segregation energy / embrittling energy and effect of Mg is not clear. This section needs to be improved.

Maybe the reviewer has not understood the text, but there is something unclear in “Upon integration of the H concentrations over all possible sites in the unit area, a total embrittling energy of $\sim 0.6 \text{ J/m}^2$ for this GB is obtained. This value is substantially more positive (i.e., more detrimental) than the embrittling energy determined for the Zr dispersoids ($\sim 2 \text{ J/m}^2$). It is also larger than for the S phase ($\sim 129 \text{ mJ/m}^2$),...”

Answer: We cordially thank the reviewer for this comment and apologize that “m” was missing for the Zr-dispersoids in the original manuscript and the embrittling energy for the Zr-dispersoids is $\sim 2 \text{ mJ/m}^2$ indeed. We changed this in the modified manuscript.

Page 8: “When distributing H atoms according to their nominal solubility over all these possible sites in a unit area of the GB, weighted by their respective segregation energy (Fig. 4C), the total contribution to the embrittling energy adds up to 600 mJ/m^2 for this GB. This value is substantially more positive (i.e., more detrimental) than the embrittling energy determined for the Al_3Zr -dispersoids (2 mJ/m^2) and the S-phase (129 mJ/m^2).”

- Suggested improvements: experiments, data for possible revision

See previous comments:

- organization of the text
- discussion about the representativeness of the results
- section focused on embrittling/segregation energies with and without Mg

Answer: We addressed these issues in the manuscript as shown above.

- References: appropriate credit to previous work

The authors forgot to cite relevant references concerning H trapping in aluminum alloys, e.g. AA2024.

Answer: We thank the reviewer for this hint. We carefully checked the references on H trapping in Al-alloys and cited them in the manuscript.

- Clarity and context: lucidity of abstract/summary, appropriateness of abstract, introduction and conclusions

The context is clearly described. The conclusions and suggestions for future work are very interesting. However, the conclusions concerning H ingress should be changed, i.e. "Al-alloys can absorb H during processing....leads to H ingress": the authors evidenced the presence of residual H in the Al alloys, but did not study the mechanisms. The comment concerning the influence of temperature is speculative, based on reference to literature, but this has not been studied in the paper.

Answer: We greatly appreciate the very positive comment from the reviewer and thank the reviewer for the pertinent suggestion on the mechanism regarding H ingress. We fully comply and removed this part from the discussion.

Referee #3 (Remarks to the Author):

Opening comment: The authors provide atom probe tomography (APT) measurements of trapping sites in a 7xxx-Al high-strength alloy, in an attempt to establish a mechanistic understanding of H-embrittlement in such alloys. This has been particularly a challenging topic for the community due to the low solubility of H in this alloy as well as the experimental difficulty in achieving spatially-resolved characterization of H at specific microstructural features. The authors also couple the experimental work with DFT calculations to explain the enrichment of H observed in intermetallic phases as well as compute the embrittling energy associated with H.

The work is of high quality and generally speaking the paper can be easily understood by experts in the field. The general notion of H enrichment at defects and grain boundaries is not necessary new from a theoretical point of view and it has been well accepted. But the work presented clearly provides direct evidence of this. Perhaps the real major conclude of this work is that insights the experiments and the simulations suggest that "when combined with Mg, the strong driving force for H to segregate to a free surface with respect to a possible interstitial site at GBs in presence of Mg, favors GB decohesion and drives the system towards

crack formation." However, I question whether this work is general enough for the general readership of Nature.

We cordially thank the reviewer for the strong support and the very positive comments. We highly appreciate that the reviewer acknowledges the challenge and difficulty of the experiment. Our manuscript is indeed the first direct near-atomic observation of H in Al-alloys and we would like to emphasize its significance by providing a mechanistic understanding of H embrittlement in Al alloys, as also clearly stated by the other two reviewers. The low solubility of H in Al-alloys makes these studies extremely challenging and thus, understanding the H-microstructure interactions has been remained elusive so far. In a period where high-strength Al alloys are expected to substantially gain momentum as key materials for weight reduction of automobiles, these problems must be better understood, to maintain safety of passengers and structures.

We agree that the enrichment of H at defects is not a new concept. A number of previous studies recognize that H may be trapped at lattice defects such as dislocations and vacancies¹⁷⁻¹⁹, yet little work has been done to investigate the effects of microstructural trapping on H segregation and transport in Al-alloys at relevant scales and at the specific microstructures¹⁹. Special about our work is the comparison of different microstructure features, namely the intermetallic phases (e.g. S phase), and Al₃Zr-dispersoids as well as GBs. It is a basis for an observation, which we believe is of general interest well beyond the specific material system: While some microstructure features (like the second-phase particles in our case) give rise to substantial trapping of H, they do not need to be sensitive for decohesion and embrittlement. Other microstructure features (like GBs) might give rise to a much lower enrichment by H, but can be critical for H embrittlement. We prove that such a criticality of H can be caused by the combination with other alloying elements (like Mg in our case) as already highlighted by the referee. Only by combining experiment and theory we were, therefore, able to demonstrate that such a combined impact can be much more severe than the segregation of H to certain defects alone. Hence, we believe that our combined experimental and modeling approach presents a significant advance in the understanding of H-assisted embrittlement in engineering Al-alloys. We expect that the mechanisms revealed here also affect other Al-alloy systems. Also, this kind of joint atomic-scale experiment and atomic-scale simulation might serve as a blueprint on how to tackle such problems in other material systems.

For this reason, we believe that the work is suited for the general readership of *Nature*.

Additional points to be addressed:

- 1) Figure 1C, the abbreviation PFZs was never defined. Also not clear how the PFZs are of relevant to the current study so what is the emphasis of Fig 1C?

Answer: We are grateful for this hint. The definition of PFZs has now been added to the caption of Fig. 1 on page 4. "PFZs: precipitate free zones".

We thank the reviewer for the question and the suggestion regarding the relevance of PFZs to the current study. As shown in Fig. 1C in the original manuscript, precipitate free zones (PFZs) next to grain boundaries (GBs) are a very characteristic and important feature of precipitation strengthened Al-alloys. These regions form from the accelerated GB precipitation behavior and the annihilation of trapped vacancies^{20,21}. This effect results in a comparatively soft zone adjacent to GBs²², which severely affects the mechanical properties and damage initiation kinetics (region of high mechanical contrast) and corrosion behavior (region of micro-/nano-galvanic elements). Hence it is very important to observe the H distribution at GBs and also in the regions adjacent to them (i.e. PFZs) and therefore corresponding results are shown in Fig. 3 of the manuscript. We indeed observe H enrichment at GBs but no clear H enrichment in the adjacent PFZs, an effect which even amplifies the contrast and vulnerability of these regions. We added the following sentence on the relevance of the PFZs to the current study on Page 3, modified Fig. 3 to include PFZs, and add the corresponding discussion items about the H distribution in the PFZs on Page 6.

Page 3: "Fig. 1D displays the typical distribution of fine precipitates in the bulk, coarse precipitates at GBs and precipitate free zones (PFZs) adjacent to GBs. Intermetallic phases (e.g. S-phase) and Al₃Zr-dispersoids that act as grain refiners are also shown."

Page 6: "The peak aged sample contains 5 nm-(Mg,Zn)-rich strengthening precipitates in the bulk and coarse 20–50 nm-sized precipitates at the GB²⁰, as well as typical PFZs adjacent to the GB (Fig. 3A)."

Page 6: "The locally increased content of D(H₂⁺) implies that the solute-decorated GB (i.e. devoid of precipitates) acts as a trap for H, while no enrichment in H and D is observed in the

adjacent PFZs (i.e. regions next to the GB), an effect which amplifies the mechanical and electrochemical contrast in these regions.

Fig. 3. APT analysis of a D-charged peak aged Al-Zn-Mg-Cu sample containing a GB (120 °C/24h). (A) The iso-surfaces highlight the dispersion of fine (Mg,Zn)-rich precipitates in the matrix, coarse ones at the GB, and Al₃Zr-dispersoids. (B) Atom maps of H and D(H₂⁺). (C) Solute distribution at the GB plane. (D) Composition profile across one Al₃Zr-dispersoid at the GB. (E) Composition profile of one (Mg,Zn)-rich precipitate at the GB. (F) Solute composition profile across the GB in between of precipitates. GBPs: grain boundary precipitates; GB: grain boundary; PFZs: precipitate free zones.

2) The authors discuss that H has low solubility in Al. However the focus of the paper is on H enrichment at bulk features. No discussion is made to explain to the reader how H would diffuse to these features at all with the exception of high-temperature processing towards the end of the manuscript. However, how H would ingress in samples present in "aqueous or humid environments" has never been discussed given the low solubility of H at room temperature.

Answer: We would like to thank the reviewer for this comment.

The low solubility of H in Al¹⁹ is an average value from bulk-Al. Yet crystal defects can assist with H absorption and act as trapping sites, as we observe here for intermetallic phases. It is expected that the H solubility at the intermetallic and GBs will exceed the true bulk solubility^{19,23,24}. Al-alloys are reactive towards H production when exposed to aqueous or humid environments^{24,25}, despite their low H solubility compared with other solid FCC materials. H-uptake is favored when atomic H is produced either chemically or electrochemically on Al surfaces²⁴. This effect becomes more pronounced if mechanical loading is applied during the H-exposure, which will promote H uptake into the specimen²⁶.

We evidenced also the presence of residual H in the Al alloys, but did not study the mechanisms on H ingress. The kinetics of H adsorption is a study on its own, which has also been done for other material systems^{19,24}. Therefore, we do not include the discussion on it as suggested by Reviewer 2.

3) The discussion about the comparison between the uncharged specimens and charged specimens is not really clear. Looking at Fig S2 it seems that the H content in Zr-dispersoids is the same for the charged and uncharged samples. So what is the conclusion here? The same is for charged and uncharged S phase. I think this discussion needs to be fully revised to make it clear to the reader what is different in both cases! Additionally, no discussion or results on comparison between H content in grain boundaries of charged and uncharged samples. It seems that this would be an important case to show/discuss. After all the main conclusion is about grain boundary embitterment. Maybe having a table that summarizes the comparisons for the H content in each defect in the charged and uncharged states would also help the reader.

Answer: We thank the reviewer for raising this question and for the helpful comment on how to improve this passage of the paper. We addressed this very point above in the reply to Review #1. We also provided a table with all the data combined to facilitate the comparison.

Table S1

H and D composition within the second phases and grain boundaries in the charged and uncharged states

Interface	State	H (at.%)	H ²⁺ /D (at.%)	C _H /C _D
Zr-dispersoid	uncharged	8.50	0	0
	D-charged	9.50	2.80	3.39
S-phase	uncharged	6.82	0.03	227
	D-charged	4.20	0.12	35
Grain boundary	uncharged	1.37	0.10	13.7
	D-charged	2.42	0.31	7.81

4) Minor point: No discussion about fig 3D?

Answer: We would like to thank the reviewer for this question. We add the discussion on the Fig. 3D to Page 6.

"Al₃Zr-dispersoids at the GB (Fig. 3D) contain 11 at.% H and 0.6 at.% D, i.e. a lower D content compared to the Al₃Zr-dispersoids in the bulk (Fig. 2B)."

5) In the conclusion the authors state "The results shed light on possible approaches for the further control and design of precipitations and GB segregation to render an improved resistance to H-induced damage." ... This seems like a general open-ended statement. Point out to the reader what are such possible approaches based on the current results.

Answer: We thank the reviewer for this suggestion. We have modified this part and added more concrete suggestions.

Page 10: “Generally, avoiding the ingress of H in the first place is extremely unlikely to work, and the best approach to mitigate H-embrittlement is therefore to control its trapping in order to maximise the components’ lifetime in-service. Our results provide indications of H-trapping sites and their respective propensity to initiate damage in the environmentally-assisted degradation, thus contributing to establish a mechanistic understanding of H-embrittlement in Al-alloys. Which specific measures can be explored on the basis of this study, to enhance the resistance of such alloys to H-induced damage, thus improving the lifetime and sustainability of high-strength lightweight engineering components? The results on the high H-enrichment in the second-phase particles provide a potential mitigation strategy for improving H-embrittlement, namely, through the introduction and manipulation of the volume fraction, shape, dispersion and chemical composition of second-phases, despite their potentially harmful effects on mechanical properties. Other strategies could aim at designing and controlling GB segregation, for instance with the goal of eliminating Mg decoration of GBs by trapping it into precipitates and keeping it in bulk solution. A third and more general avenue against environmental degradation lies in reducing the size of the precipitation free zones in these alloys, with the goal to mitigate the H-enhanced contrast in mechanical and electrochemical response between the H-decorated GBs and the less H-affected adjacent regions.

References

- 1 Chen, Y. S. *et al.* Direct observation of individual hydrogen atoms at trapping sites in a ferritic steel. *Science* **355**, 1196-1199 (2017).
- 2 Chen, Y.-S. *et al.* Observation of hydrogen trapping at dislocations, grain boundaries, and precipitates. *Science* **367**, 171-175 (2020).
- 3 Breen, A. J. *et al.* Solute hydrogen and deuterium observed at the near atomic scale in high-strength steel. *Acta Mater.* **188**, 108-120 (2020).
- 4 Gemma, R., Al-Kassab, T., Kirchheim, R. & Pundt, A. APT analyses of deuterium-loaded Fe/V multi-layered films. *Ultramicroscopy* **109**, 631-636 (2009).
- 5 Takahashi, J., Kawakami, K. & Tarui, T. Direct observation of hydrogen-trapping sites in vanadium carbide precipitation steel by atom probe tomography. *Scr. Mater.* **67**, 213-216 (2012).
- 6 Nie, J. F. & Muddle, B. C. Strengthening of an Al–Cu–Sn alloy by deformation-resistant precipitate plates. *Acta Mater.* **56**, 3490-3501 (2008).
- 7 Chao, P. & Karnesky, R. A. Hydrogen isotope trapping in Al–Cu binary alloys. *Mater. Sci. Eng. A* **658**, 422-428 (2016).
- 8 Kamoutsi, H., Haidemenopoulos, G. N., Bontozoglou, V. & Pantelakis, S. Corrosion-induced hydrogen embrittlement in aluminum alloy 2024. *Corros. Sci.* **48**, 1209-1224, (2006).
- 9 Charitidou, E., Papapolymerou, G., Haidemenopoulos, G., Hasiotis, N. & Bontozoglou, V. Characterization of trapped hydrogen in exfoliation corroded aluminium alloy 2024. *Scr. Mater.* **41**, 1327-1332 (1999).
- 10 Larignon, C. *et al.* Investigation of Kelvin probe force microscopy efficiency for the detection of hydrogen ingress by cathodic charging in an aluminium alloy. *Scr. Mater.* **68**, 479-482, (2013).
- 11 Saitoh, H., Iijima, Y. & Hirano, K. Behaviour of hydrogen in pure aluminium, Al-4 mass% Cu and Al-1 mass% Mg₂Si alloys studied by tritium electron microautoradiography. *J. Mater. Sci.* **29**, 5739-5744 (1994).
- 12 Sun, B., Krieger, W., Rohwerder, M., Ponge, D. & Raabe, D. Dependence of hydrogen embrittlement mechanisms on microstructure-driven hydrogen distribution in medium Mn steels. *Acta Mater.* **183**, 313-328 (2020).

- 13 Chang, Y. *et al.* Ti and its alloys as examples of cryogenic focused ion beam milling of environmentally-sensitive materials. *Nat. Commun.* **10**, 942 (2019).
- 14 https://ec.europa.eu/clima/policies/transport/vehicles/regulation_en#tab-0-2.
- 15 Chen, J. H., Costan, E., van Huis, M. A., Xu, Q. & Zandbergen, H. W. Atomic Pillar-Based Nanoprecipitates Strengthen AlMgSi Alloys. *Science* **312**, 416-419 (2006).
- 16 Sun, W. *et al.* Precipitation strengthening of aluminum alloys by room-temperature cyclic plasticity. *Science* **363**, 972-975 (2019).
- 17 Ishikawa, T. & McLellan, R. B. The diffusivity of hydrogen in aluminum. *Acta Metall.* **34**, 1091-1095, (1986).
- 18 Itoh, G., Koyama, K. & Kanno, M. Evidence for the transport of impurity hydrogen with gliding dislocations in aluminum. *Scr. Mater.* **35**, 695-698, (1996).
- 19 Young, G. A. & Scully, J. R. The diffusion and trapping of hydrogen in high purity aluminum. *Acta Mater.* **46**, 6337-6349, (1998).
- 20 Zhao, H. *et al.* Segregation assisted grain boundary precipitation in a model Al-Zn-Mg-Cu alloy. *Acta Mater.* **156**, 318-329 (2018).
- 21 Unwin, P., Lorimer, G. & Nicholson, R. The origin of the grain boundary precipitate free zone. *Acta Metall.* **17**, 1363-1377 (1969).
- 22 Vasudevan, A. K. & Doherty, R. Grain boundary ductile fracture in precipitation hardened aluminum alloys. *Acta metall.* **35**, 1193-1219 (1987).
- 23 Edwards, R. A. H. & Eichenauer, W. Reversible hydrogen trapping at grain boundaries in superpure aluminium. *Scr. Metall.* **14**, 971-973 (1980).
- 24 Scully, J. R., Young, G. A. & Smith, S. W. Hydrogen Embrittlement of aluminum and aluminum-based alloys, in: R.P. Gangloff, B.P. Somerday (Eds.), *Gaseous Hydrog. Embrittlement Mater. Energy Technol.*, Woodhead Publishing, **2**, 707-768 (2012).
- 25 Su, H. *et al.* Assessment of hydrogen embrittlement via image-based techniques in Al-Zn-Mg-Cu aluminum alloys. *Acta Mater.* **176**, 96-108, (2019).
- 26 Scamans, G., Alani, R. & Swann, P. Pre-exposure embrittlement and stress corrosion failure in AlZnMg Alloys. *Corros. Sci.* **16**, 443-459 (1976).

Reviewer Reports on the First Revision:

Referee #1:

Remarks to the Author:

The authors have done a reasonable job of addressing all the little things (terminology, fixing incorrect statements, tensile curves, etc...) I have highlighted in my original review, but the major important question of the generality of the conclusion regarding co-segregation of H and Mg causing embrittlement was not addressed.

In this reviewer's view, the measurements of H at dispersoids, precipitates, is fully expected based on everything already known regarding H effects in engineering alloys. The potentially major conclusion is that the first order effect of the embrittlement is due to the co-seg of H and Mg. I note that Reviewer #3 also shares this view that this is the interesting part of the article. If this finding can be generalised in some way (ie. it is relevant to more than this particular alloy) then it may be significant enough and general enough for a journal like Nature. But the authors decline the chance to investigate this in Mg-free alloys to show the generalization. If they do not like Al-Cu, then try something else, Al-Zn etc....

As the manuscript is written, they cannot generalise the claim regarding H and Mg co-seg causing the embrittlement beyond this particular alloy. If there is no possibility of generalization, how can this be published in a general journal? The work could be published but Nature Comms would be more appropriate.

Referee #2:

Remarks to the Author:

This manuscript corresponds to a revised version of an original manuscript. The authors carefully took into account the revisions suggested by the reviewer. Therefore, the reviewer

thinks that the manuscript is suitable for publication in the present form.

Summary of the key results

This paper provides identification of efficient hydrogen trapping sites on the basis of quantitative analyses, which will be helpful to design microstructures with a lower susceptibility to hydrogen embrittlement.

Originality and significance: if not novel, please include reference

The paper is original and of great significance. It is of major interest in the current context.

Data & methodology: validity of approach, quality of data, quality of presentation

There is no problem concerning the data and methodology. The presentation of the manuscript has been improved.

Appropriate use of statistics and treatment of uncertainties

The authors addressed in the revised version of the manuscript the issue related to the representativeness of their analyses. The reviewer thanks them for that.

Conclusions: robustness, validity, reliability

The robustness of the work is now clear. The reviewer has appreciated the analyses performed inside a crack.

Suggested improvements: experiments, data for possible revision

The manuscript is suitable for publication in the present form.

References: appropriate credit to previous work

Now, it is correct.

Clarity and context: lucidity of abstract/summary, appropriateness of abstract, introduction and conclusions

The context is clearly described. The conclusions and suggestions for future work are very interesting.

Referee #3:

Remarks to the Author:

The authors provided a detailed response to all the questions raised and this reviewer is satisfied with those responses.

Author Rebuttals to First Revision:

Referee #1 (Remarks to the Author):

The authors have done a reasonable job of addressing all the little things (terminology, fixing incorrect statements, tensile curves, etc...) I have highlighted in my original review, but the major important question of the generality of the conclusion regarding co-segregation of H and Mg causing embrittlement was not addressed.

In this reviewer's view, the measurements of H at dispersoids, precipitates, is fully expected based on everything already known regarding H effects in engineering alloys. The potentially major conclusion is that the first order effect of the embrittlement is due to the co-seg of H and Mg. I note that Reviewer #3 also shares this view that this is the interesting part of the article. If this finding can be generalised in some way (ie. it is relevant to more than this particular alloy) then it may be significant enough and general enough for a journal like Nature. But the authors decline the chance to investigate this in Mg-free alloys to show the generalization. If they do not like Al-Cu, then try something else, Al-Zn etc...

As the manuscript is written, they cannot generalise the claim regarding H and Mg co-seg causing the embrittlement beyond this particular alloy. If there is no possibility of generalization, how can this be published in a general journal? The work could be published but Nature Comms would be more appropriate.

Answer: We are grateful for these hints and appreciate that the reviewer acknowledges the significant contribution of this work to understanding the embrittlement effect at grain boundaries (GBs) in Al alloys. We thank the reviewer for his / her pertinent suggestion regarding conducting additional experiments, i.e. testing Mg-free Al alloys to show the generalization of this work. We fully comply and have therefore now conducted additional experiments on an Al-Zn alloy. More specific, an Al-2.31 at. % Zn alloy was used, which contains a similar amount of Zn as in the studied Al-Zn-Mg-Cu alloy (2.69 at.%), yet, is devoid of Mg as suggested by the reviewer. Table 1 shows the chemical composition of the Al-Zn alloy obtained from wet chemical analysis. The cast ingot was homogenized at 360 °C for 6 h and water quenched, followed by hot rolling at 345 °C from 20 to 3 mm thickness and solution treatment at 360 °C for 1 hour and a final quench in water.

Table 1. Material composition

Alloy	Zn	Fe	Si	Al
(wt.%)	5.41	0.15	0.02	balance
(at.%)	2.31	0.08	0.02	balance

We conducted Hydrogen (H)-charging for the Al-Zn alloy using the same H-charging method for the Al-Zn-Mg-Cu material, as shown in the supplemental material part (Page 14) in the manuscript. Tensile experiments were then immediately done on the H-charged Al-Zn samples. The results can be seen below. We reveal that the uncharged and H-charged samples show similar mechanical properties. Note that a significant embrittling effect with 51.5% elongation loss is observed in the H-charged Al-Zn-Mg-Cu samples as compared with the uncharged samples (Fig. 1(A)).

Fig. R1. Engineering stress-strain curves of uncharged and H-charged Al-Zn samples.

We also investigated the fracture mode of the H-charged and uncharged Al-Zn samples. Typical scanning electron micrographs of the fracture surfaces are displayed below. The uncharged and H-charged samples both show transgranular ductile, dimple-type fracture

surfaces (Fig. R2A-B). In contrast, a predominantly intergranular fracture driven by the crack formation at GBs is observed in the H-charged Al-Zn-Mg-Cu alloy, while a dimple-type fracture surface is shown in the uncharged samples.

Fig. R2. Typical scanning electron microscopy fractography of the Al-Zn samples: (A) uncharged; (B) H-charged.

We add these items as additional information to Page 10 in the modified manuscript. These additional results clearly confirm our claim, as the Zn doped samples do not show any sign of H-embrittlement, neither in the tensile test results, nor in the fracture metallography.

“Further investigation on the elemental distribution at a H-induced intergranular crack using scanning Auger electron microscopy reveals the enrichment of Mg at the cracked GB (Extended Data Fig. 9). The enrichment is even stronger (by a factor of 2) than the Mg concentration at the GB (Fig. 3f). To confirm the generality of the role of Mg we also show that no H-embrittlement features occurred in a Mg-free Al-5.41 wt.% Zn alloy that was used as reference material. The alloy was exposed to the same H-charging and tensile test conditions, but no sign of H-embrittlement was found, neither in the tensile test results nor in the metallographic fractography (Extended Data Fig. 10). These findings support the result that the co-segregation of Mg and H to free surfaces provides the driving force for the embrittlement of GBs.”

We believe that with these additional experiments and very clear results the reviewer's concern has been fully addressed. We will be ready to make further efforts to improve the quality of the manuscript, should further points arise.

Referee #2 (Remarks to the Author):

This manuscript corresponds to a revised version of an original manuscript. The authors carefully took into account the revisions suggested by the reviewer. Therefore, the reviewer thinks that the manuscript is suitable for publication in the present form.

- Summary of the key results

This paper provides identification of efficient hydrogen trapping sites on the basis of quantitative analyses, which will be helpful to design microstructures with a lower susceptibility to hydrogen embrittlement.

- Originality and significance: if not novel, please include reference

The paper is original and of great significance. It is of major interest in the current context.

- Data & methodology: validity of approach, quality of data, quality of presentation

There is no problem concerning the data and methodology. The presentation of the manuscript has been improved.

- Appropriate use of statistics and treatment of uncertainties

The authors addressed in the revised version of the manuscript the issue related to the representativeness of their analyses. The reviewer thanks them for that.

- Conclusions: robustness, validity, reliability

The robustness of the work is now clear. The reviewer has appreciated the analyses performed inside a crack.

- Suggested improvements: experiments, data for possible revision

The manuscript is suitable for publication in the present form.

- References: appropriate credit to previous work

Now, it is correct.

- Clarity and context: lucidity of abstract/summary, appropriateness of abstract, introduction and conclusions

The context is clearly described. The conclusions and suggestions for future work are very interesting.

We very cordially thank the reviewer for the strong support and the helpful comments.

Referee #3 (Remarks to the Author):

The authors provided a detailed response to all the questions raised and this reviewer is satisfied with those responses.

We are most grateful for the support and comments from the reviewer.

fReviewer Reports on the Second Revision:

Referee #1:

Remarks to the Author:

The experiments on Al-Zn are welcomed and help make the case that the authors observations have some generality. Of course, they will be criticised in the literature because the yield strength of the Al-Zn alloys is an order of magnitude lower than the Al-Zn-Mg-Cu they use as the main material, and the susceptibility to H-embrittlement is well known to depend on the yield strength of the alloys. This is why Al-Cu would have been a better choice (it can be age hardened to at least 300MPa) and still contains no Mg.

But the authors have shown some proof of generality and that is what is required, in my view, for publication.